

**Detection of dietary stress and geophagic behaviour forced by dry seasons in Miocene _Gomphotherium_**

Rute Coimbra [(a)], Niels de Winter [(b,c)], Maria Ríos [(d, e)], Rui Bernardino [(f)], Darío Estraviz-López [(d, e)], Priscila Lohmann [(d)], Roberta Martino [(d, e)], Aurora Grandal-d'Anglade [(g)], Fernando Rocha [(a)], Philippe Claeys [(c)]

(a) Geobiotec, Dpt. of Geosciences, University of Aveiro, Portugal

(b) Dept. of Earth Sciences, Vrije Universiteit Amsterdam, Netherlands

(c) Archaeology, Environmental Changes & Geo-chemistry group, Vrije Universiteit Brussel, Belgium

(d) Dpt. of Earth Sciences, NOVA School of Science and Technology, Universidade Nova de Lisboa, GeoBioTec, Caparica, Portugal

(e) Museu da Lourinhã, Lourinhã, Portugal

(f) Jardim Zoológico de Lisboa, Lisboa, Portugal

(g) ESCI, Instituto Universitario de Xeoloxía, Universidade da Coruña, A Coruña, Spain

**Corresponding author and first author:**

Rute Coimbra rcoimbra@ua.pt

**Abstract**

To access the impact of anthropogenic emissions and land use change on Earth's climate and biodiversity, studies into the environment and ecology of natural ecosystems during past warm periods are required. The Miocene Climatic Optimum is a key reference period for future global warming scenarios. However, studies uncovering Miocene climate have heavily favoured marine environments, leaving the impact of warming on terrestrial ecosystems understudied. Here, we present a multidisciplinary study into the chemical composition of fossil _Gomphotherium angustidens_ (Proboscidea, Mammalia) teeth from the Middle Miocene Vb division (~15.9–16.1Ma) of western Portugal (Chelas Valley, Lisbon, Lusitanian basin) and their sedimentological context. Trace element and stable isotope compositions in these fossil teeth are compared with similar measurements in molars of a taxonomically related modern African elephant (captive _Loxodonta africana_). Results reveal seasonal-scale variability in trace elements in both fossil and modern proboscidean tooth enamel, which are interpreted as evidence for seasonal changes in diet. Periodic increases in Na, Fe and Si in _G. angustidens_ demonstrate intake of sediment in the diet during fixed times of the year, a behaviour type previously described in modern elephants during dry seasons. In combination with the heavier carbon and oxygen isotopic composition in _G. angustidens_ compared to _L. africana_, the terrestrial climate in Miocene Portugal appears characterized by seasonally dry periods, which forced geophagy behaviour of these large mammals and likely had significant consequences for the composition of Miocene ecosystems (e.g., food/water availability and potential seasonal range shifts) in southwestern Europe.



## 1. Introduction

### 1.1 The Middle Miocene as a reference period for warm climate

The study of historical variations in climate and environment has emerged as a potent method for understanding the scale, duration, and trajectory of global change. This approach also aids in assessing and forecasting potential outcomes in future scenarios, as underscored by the Intergovernmental Panel on Climate Change (IPCC) in its 6th assessment report (IPCC, 2023). The Middle Miocene Climate Optimum (MCO; 17 – 15 Ma) represents a period of major global warming within the Cenozoic cooling trend (Domingo et al., 2009; Harzhauser et al., 2011; Meckler et al., 2022; Westerhold et al., 2020). This brief hot period, reaching ~ 600 ppm atmospheric $CO_2$, closely mimics projections for future temperatures under a moderate IPCC warming (Meinshausen et al., 2020; Methner et al., 2020; Super et al., 2018). Therefore, MCO constitutes an appropriate analogue to access the predictive nature of climate models (Burls et al., 2021; Goldner et al., 2014; Holbourn et al., 2015; Steinthorsdottir et al., 2020).

Employing biogeochemical techniques to extract short-term geochemical information from fossil skeletal remains constitutes a highly informative method, as it enables the characterization of past global changes for a timespan that far exceeds instrumental records (Meckler et al., 2022; Westerhold et al., 2020; Zachos et al., 2001, 2008). A deeper exploration of the marine geological record has been carried out from a biogeochemical perspective, specifically utilizing stable oxygen isotope ratios ($\delta^{18}O$) as proxies for past ocean water density (salinity and/or temperature). The $\delta^{18}O$ proxy has been primarily used in the study of marine paleoarchives (Westerhold et al., 2020; Zachos et al., 2008). Since a similarly detailed approach is missing from the continental realm, it is crucial to enhance biogeochemical investigations into proxies reflecting terrestrial records to achieve a comprehensive understanding of the climatic, environmental, and ecological changes that have shaped the planet.

### 1.2 Bioapatite as an environmental archive

Bioapatite ($Ca_{10}(PO_4, CO_3)_6(OH, F)$; LeGeros, 1986), the material from which teeth and bones in terrestrial vertebrates are made, constitutes a promising archive for recording short-term environmental variability in the terrestrial realm. Bioapatite is one of the strongest biogenic materials and has excellent fossilization potential (Lee-Thorp and Sponheimer, 2003). The oxygen isotope composition of carbonate ($\delta^{18}O_c$) structurally bound to bioapatite in skeletal tissues primarily reflects the climatic and environmental conditions experienced by the examined individual during its life (Cerling et al., 1997a; de Winter and Claeys, 2016; Fricke and O'Neil, 1996). For terrestrial vertebrates, these measurements mirror the $\delta^{18}O$ content of body water, which, in turn, captures the $\delta^{18}O$ composition of meteoric water after correcting for metabolic fractionation, and are related to the mineralization temperature of the mineral (Ayliffe et al., 1992). Furthermore, with increasing aridity, the enrichment of $^{18}O$ in meteoric and vegetation water, and therefore in tooth enamel, intensifies due to evaporation effects- Rayleigh fractionation (Tütken et al., 2007). Hence, the bioapatite $\delta^{18}O_c$ signature found in fossil vertebrates serves as a valuable source of information about temperature and aridity levels within terrestrial environments (Koch et al., 2007).

Initial investigations into the carbon isotope value of bioapatite ($\delta^{13}C_{ap}$) in terrestrial vertebrates revealed its capacity to document dietary choices, facilitating the reconstruction of habitats (such as forested versus open areas). In the context of herbivore species, $\delta^{13}C_{ap}$ values are influenced by the photosynthetic pathway of consumed plants, enabling differentiation among C3, C4, and Crassulacean Acid Metabolism (CAM) metabolic pathways (Koch et al., 2007).

The recent development of high-resolution micro-XRF (μXRF) line scanning to analyze trace element abundances on cleaned surfaces of mammal molars serves as a valuable complement to conventional isotope proxies (de Winter et al., 2019; de Winter and Claeys, 2016). The findings demonstrate a link between seasonal fluctuations and trace element patterns in enamel, namely in Sr/Ca, Zn/Ca, K/Ca, Fe/Ca and S/Ca. These ratios reflect the intake of trace elements through dust (discerning between summer and winter) or dietary modifications. The documented connection constitutes a promising



avenue for utilizing these trace element ratios as a novel proxy to explore seasonal fluctuations in both
the ancient environment and dietary habits of extinct Proboscideans, potentially applicable to other
mammalian species (de Winter et al., 2017; Kohn and Cerling, 2002) .

1.3 Proboscidean teeth as archives for terrestrial environmental variability and paleobiology
Mammal tooth enamel has emerged as a valuable recorder of paleo-seasonality due to its resistance
to diagenesis and its sequential growth, facilitating the retrieval of records with exceptional temporal
resolution (Blaise and Balasse, 2011). An additional benefit of employing mammal teeth for sub-annual
environmental reconstructions is the potential to amalgamate several teeth from a single individual,
forming a composite time series. This combination enables the generation of extended, uninterrupted
records depicting fluctuations in paleoenvironmental conditions throughout the years when the teeth
underwent mineralization (De Winter et al., 2016; Kohn and Cerling, 2002).
Proboscideans, which are among the largest land herbivores of the Neogene era (Fig. 1A), are
polyphyodonts, undergoing cycles of tooth replacement over their lifetimes (Lee et al., 2012). New
teeth gradually develop at the rear of the mouth, moving forward to displace and replace the anteriorly
located older ones (molar progression; Fig. 1B) (Lee et al., 2012; Metcalfe and Longstaffe, 2012). Their
molars are lophodont (Fig. 1C and D), composed of enamel lophs filled with dentin and surrounded by
cementum (Lee et al., 2012). Throughout the process of tooth formation, the growth of enamel initiates
at the front of the crown and advances in a loph-by-loph manner towards the distal end of the tooth
and the root. This progression continues until it reaches the point where the crown and root meet,
offering a distinct pathway for conducting analyses of seasonality (Fig. 1). The rates at which enamel
accumulates in proboscidean molars suggest the possibility of capturing an environmental record
spanning up to 15 years (Metcalfe and Longstaffe, 2012), with an enamel accretion rate up to
~13 mm/yr (Dirks et al., 2012; Esker et al., 2019; Kowalik et al., 2023). Proboscideans possess a diverse
evolutionary lineage and extensive fossil record, rendering them a captivating subject of study. This
characteristic offers the potential for terrestrial paleoclimate reconstruction, spanning from the Late
Paleocene era to the present day (Cantalapiedra et al., 2021; Liu et al., 2008).

1.4 *G. angustidens* and modern elephant evolution and biogeography
*Gomphotherium* (Fig. 1A) is an extinct genus of gomphothere proboscidean, which once roamed the
Neogene landscapes of Eurasia, Africa, and North America (Wang et al., 2014). Its origin can be traced
back to Africa during the late Oligocene to early Miocene period (Wang et al., 2017). The earliest
remnants of *Gomphotherium*, date back approximately 19.5 million years, and were discovered in
Africa (Wang et al., 2017). Around 19 million years ago, *Gomphotherium* embarked on a range shift into
Eurasia via the "*Gomphotherium* Land Bridge", a land bridge that connected Eurasia to Afro-Arabia
(Harzhauser et al., 2007). Upon its arrival in Eurasia, *Gomphotherium* experienced rapid evolutionary
changes, reaching its zenith of diversity during the Early-Middle Miocene epoch. As the Pliocene
dawned, the last known *Gomphotherium* species vanished from North America, around 5 million years
ago (MacFadden et al., 2015).
Most *Gomphotherium* species reached a size comparable to the modern Asian elephant. They were
characterized by their distinctive long lower jaw tusks and four-tusked dentition (Larramendi, 2015).
Gomphotheres inhabited a diverse range of habitats, and while the majority of *Gomphotherium* species
are believed to have been browsers, indications suggest that certain specimens of *G. steinheimense*
from China showed grazing habits (Wu et al., 2018). Modern elephants are primarily considered
browsers, rather than grazers, although they do consume grasses as part of their diet, especially during
times of food scarcity (Wu et al., 2018).



The evolution of elephantids is believed to have stemmed from gomphotheres, and the forebears of
modern elephants might be African members of the "tetralophodont gomphothere" *Tetralophodon*
(Geraads et al., 2019). The family's earliest members date back to the Late Miocene period,
approximately 9-10 million years ago (Saegusa et al., 2014). While early members of Elephantidae had
lower tusks, these were lost in later members as the feeding function of the mandible shifted gradually
to the trunk, possibly as an adaptation to slightly more grazing habits (Li et al., 2023; Mothé et al.,
2016). By the end of the Miocene epoch, elephants and mammoths diverged from each other, and
during the Pliocene the elephantid range shift from Africa started, leading to the arrival of mammoths
and *Elephas* in Eurasia approximately 3.8-3 million years ago (Iannucci and Sardella, 2023).
One critical aspect of proboscidean evolution was the modification of their check teeth occlusal
morphology, which changed from the low-crowned bunolophodont molars of the trilophodont
*Gomphotherium* with few lophs, to the high-crowned and multi-plated teeth of the more derived
elephantiforms such as *Loxodonta* (Wu et al., 2018). This evolution is of relevance because such dental
change allowed them to consume more abrasive foods with the establishment of C4 vegetation
between 8 to 5 Ma (Wu et al., 2018). While these changes in the morphological and ecological context
may cause some differences in the composition of teeth from extant and the extinct specimens,
comparison between modern elephants and the Middle Miocene *Gomphotherium* still contributes to
our understanding of how modern elephants and their ancestors adapted and diversified over the last
millions of years.

1.5 *Gomphotherium* dietary and environmental changes in Middle Miocene climate
This study presents a comparison between seasonal patterns in trace element concentrations of and
extant elephants and extinct Middle Miocene *Gomphotherium*. The Miocene record in Portugal
provides favourable conditions for conducting this research on fossil *Gomphotherium* specimens
(Antunes and Pais, 1984; Pais et al., 2012), since it is supported by a robust stratigraphic framework
and detailed depositional sequences, biostratigraphy and isotopic dating based on marine fossils, while
also exhibiting a substantial connection with terrestrial sedimentary formations.
The Middle Miocene *G. angustidens* material described here was collected in Portugal (Lisbon region;
Fig. 2A B and C). The specimens lived prior to the establishment of C4 vegetation, which happened
around 8-5 million years ago (Cerling et al., 1997a), and before the emergence of Elephantidae from
the ancestral stock of Miocene gomphotheres (Cerling and Harris, 1999). This temporal division
coincides with the Middle Miocene Climatic Optimum (MMCO), which occurred approximately 14 to
17 million years ago. During this period, the Earth experienced relatively warmer and more stable
climatic conditions, with higher temperatures than immediately before or after this interval. The
MMCO was characterized by reduced polar ice and relatively uniform temperatures across latitudes,
making it a period of significant climatic stability during the Miocene epoch (Westerhold et al., 2020;
Zachos et al., 2001).
Climatic assessments based on marine samples from the Middle Miocene of the Lower Tagus Basin
(LTB) indicate prevailing tropical conditions in the studied region (Pais et al., 2012). During the Upper
Burdigalian and Langhian stages(~17 – 15 Ma), water temperatures reached a peak, resembling the
present-day climate of the Guinea Gulf (Pais et al., 2012). On land, Portuguese records indicate a
decline in both humidity and temperature, leading to the prevalence of savanna or steppe habitats
with gallery forests found alongside rivers (Pais et al., 2012). Subsequently, temperatures declined.
Evidence from continental faunas and sediments indicates a cyclical alternation of moist and arid
periods, , with the Langhian stage (~16.5 – 15 Ma) being the driest (Antunes and Pais, 1984; Pais et al.,
186  2012).






**2. Materials**


*2.1 Gomphotherium angustidens*
Insights into continental Miocene landscapes are derived from the analysis of molars, representing the
extinct proboscidean. The selected *G. angustidens* specimens were collected at Quinta da Farinheira,
Chelas Valley (Fig. 2B), falling within the Cotter (1956) subdivision Vb, and the subdivision SDL1 (
(~15.9–16.1 million years ago (Antunes and Ginsburg, 2003). It belongs to the Middle Miocene
Langhian stage and the MN5 biozone and the final part of MN4 (13.7–16.0 Ma; Fig. 2C). The encasing
friable reddish terrigenous material presents reddish hues (Fig. 2D and E) provides ideal conditions to
extract the molars of interest without risking mechanical damage. This subdivision includes the
Burdigalian and Langhian stages (Pais et al., 2012). The specimens are curated at the Department of
Earth Sciences of the Universidade NOVA de Lisboa and are molars accessioned with the numbers: FCT-
DCT-4943, FCT-DCT-4944, and FCT-DCT-4945 (Fig. 3A to C and Fig. 4).

*2.2 Loxodonta africana*
The extant proboscidean analogue material belongs to the African elephant *Loxodonta africana*. It was
provided by Jardim Zoológico de Lisboa (Lisbon Zoo) and consists of a complete molar tooth (Fig. 3D
and Fig. 4) also deposited at the Department of Earth Sciences of the Universidade NOVA de Lisboa
(FCT-DCT-4946). It belongs to an adult individual, born in the wild and later moved into an enclosed
environment in the Lisbon Zoo until its death, at the end of the 20th century. It is not uncommon to
find fallen elephant molars amongst hay and other materials during cleaning activities at the Zoo. This
is the case of the sample used in this study, accounting for the uncertainty regarding to which individual
they belonged to, as well as if the molar was mineralized in the wild or during captivity. Since a few
decades, the Lisbon Zoo shelters African elephants born and raised in the wild as well as others born
under their care, a clear reflection of their reproductive success under optimum conditions. Animal
enrichment includes (among many others) a tight control on substrate (soil) composition, requiring
periodic addition of external components from variable sources to stimulate natural behaviours.

**3. Methods**


3.1 Sample preparation
Three molars, likely of different individuals, of *G. angustidens* (FCT-DCT 4943, FCT- DCT 4944 and FCT-
DCT 4945) and one molar of *L. africanus* (FCT-DCT 4946)(examples in Fig. 3 and 4) were cut in mesio-
distal (anterior-posterior) direction, perpendicular to the orientation of the lophs in *L. africana* and the
cusp rows in *G. angustidens*, to reveal a cross section through the cusps of the molars (following Uno
et al., 2020). Cross sections were polished using $CeO_2$ powder suspension to ensure a flat surface for
µXRF scanning (Fig. 4). Sectioning through the molars, while destructive, targeted the enamel in the
cusps/lophs using µXRF line scanning to obtain time series of trace element composition through the
teeth (see **section 3.4**).

3.2 Mineralogical analysis of soil samples
To characterize the chemical components of soil potentially contributing to the composition of the
molar enamel during mineralization and/or diagenetic evolution, the mineralogical composition of soil
samples associated with both modern and fossil molars used in this study was analysed. Samples were
taken from sediment material found directly around the molars and inside the cavities between the
cusps. In samples associated with modern elephant models, soil particles were isolated from the
organic matter in material lodged between the molar cusps by soaking samples in deionized water and
decanting the liquid and floating constituents from the samples. In this way, only soil particles were



taken into account for further analysis. Sediment samples associated with fossil and modern molars
were analysed using X-Ray Diffraction (XRD) with Cu-Kα radiation, carried out using PANALYTICAL
Phillips X'Pert PW3040/60 equipment, with the X'Pert 2.0 and Profit software (Department of
Geosciences, University of Aveiro, Portugal). Scans were run between 4 and 65° 2θ for bulk samples
and between 2 and 40° 2θ of non-oriented powder mounts of fine fractions for clay mineral
identification. Divergence slit was fixed at 1.0, receiving slit at 0.20; the scan step size of 0.02 for 3050
points at continuous scan; time per step was 1 sec/2θ. Fine fraction analysis was carried out in the air-
dry state and after glycerol saturation and heat treatment (500°C). Peak identification and semi-
quantitative estimation were performed using HighScore software (Malvern Panalytical v. 4.9; ICDD
database-International Centre for Diffraction Data). Identification of the different mineral phases
followed the criteria recommended by (Brown and Brindley, 1980; Schultz, 1964) and the Joint
Committee for Powder Diffraction Standards. Semi-quantitative mineralogical analyses followed the
criteria recommended by (Mellinger, 1979; Schultz, 1964; Thorez, 1976); peak areas of the specific
reflections were calculated and weighted by empirically estimated factors, according to (Galhano et al.,
1999; Oliveira et al., 2002).

3.3 Trace element maps and profiles
3.3.1    Instrument setup
Concentrations of Ca, P, Na, Mg, Sr, S, Si, Fe, Al, K and Mn are determined across sections through
proboscidean molars using an energy-dispersive Bruker M4 Tornado µXRF scanner (Bruker nano GmbH,
Berlin, Germany) at AMGC in Brussels. The XRF spectra produced enable detection and quantification
of a wide array of elements (see de Winter and Claeys, 2016), however, the discussion is limited to the
selected elements referred above. This selection was based on the degree of significance during
elemental data processing (details in Appendix). We first mapped the entire cross section of each
specimen using semi-quantitative mapping to identify the best-preserved areas in the fossil teeth by
comparison between fossil and modern specimens. Then, we targeted those well-preserved enamel
sections using quantitative elemental profiling (examples in Fig. 5).
The Bruker M4 Tornado features a Rh-anode X-ray source operated at 50 mV and 600 µA (30 W) and
two 30 mm$^2$ silicon drift detectors mounted in a low vacuum (20 mbar) chamber under a precise
orientation such that the incoming and outgoing (detected) X-ray radiation describe a 90° angle when
they hit the sample (see de Winter and Claeys, 2016). The X-rays from the source are focused using a
poly-capillary lens. This configuration ensures that exciting X-rays can be focused on a round ~25 µm
diameter spot on the polished sample surface (calibrated for Mo-kα radiation), which is mounted on a
XYZ moving stage to allow collection of a spatial array of XRF spectra. Peaks in XRF spectra belonging
to elements of interest are identified and integrated using Bruker Esprit software, which includes a
deconvolution algorithm for estimating the relative contributions of overlapping peaks in the XRF
spectrum.

3.3.2    µXRF mapping
The entire surface of all cross sections is mapped using µXRF to produce semi-quantitative maps of
trace element distribution. For µXRF maps, the X-ray source is moved along the sample surface
continuously in a raster pattern while XRF spectra accumulate in 25 µm by 25 µm pixels with a total
integration time of 1 ms (following de Winter and Claeys, 2016). This sampling strategy does not allow
enough integration time for spectrum-by-spectrum quantification of trace element concentrations, but
instead produces semi-quantitative element distribution maps by creating false-colour images of the
deconvoluted     surface    area    under    element    peaks    (Fig.    5    and
https://doi.org/10.5281/zenodo.14882824). These maps were not used to determine accurate
concentrations of elements in the specimens, but used to guide the optimal location for quantitative
XRF line scanning.






### 3.3.3 µXRF line scanning

Quantitative µXRF trace element profiles are measured along the exposed enamel in cross sections
through the cusps and lophs of the proboscidean molars (see Fig. 3). We apply point-by-point line
scanning (following (Vansteenberge et al., 2020), allowing the detectors to count fluorescent X-rays
returning from the sample for 60 seconds per point (following de Winter and Claeys, 2016). This
approach allows enough measurement time per point to reach the Time of Stable Accuracy and Time
of Stable Reproducibility recommended in (de Winter et al., 2017), thereby achieving a compromise
between spectrum quality, measurement time and spatial resolution of the trace element profiles.
Trace element concentrations are quantified using the matrix-matched bioapatite calibration
developed in (De Winter et al., 2016) using a combination of in-house and certified bioapatite
standards. This method achieves measurement errors better than 10% for all elements considered in
this study (see detailed error analysis in de Winter and Claeys, 2016).

### 3.4 Stable isotope analyses

After carrying out the µXRF analyses, enamel samples were drilled using a hand-held dental drill
(Dremel equipped with a 3 mm drill-bit) from the cross sections through the molars for stable carbon
($\delta^{13}$C) and oxygen ($\delta^{18}$O) isotope analyses of the structurally bound carbonate in the tooth bioapatite,
performed at Universidade da Coruña (A Coruña, Spain). Bioapatite samples (N = 42) were drilled from
multiple parallel cusps and various heights along the cusps within the molars to quantify inter-tooth
variability in isotopic composition. Approximately 1 mg of enamel powder was used for stable isotope
analyses. No pre-treatment is carried out since the samples were obtained from clean cross sections
through the molars and were considered unlikely to have been in contact with diagenetic fluids
considering the taphonomic history of the sample site (see **section 2**). Therefore, the risk of introducing
diagenetic (labile) carbonate fractions in the samples was sufficiently low to motivate the choice not to
use pre-treatment techniques, which have been demonstrated to affect the original isotopic
composition of enamel in some cases (Pellegrini and Snoeck, 2016; Wood et al., 2021). The carbonate
was extracted following the protocol by(De Winter et al., 2016). Extracted carbonate samples were
reacted with 99% orthophosphoric acid ($H_3PO_4$ + $H_4P_2O_7$) in a carbonate preparation device (GasBench
II, by ThermoFinnigan) to produce $CO_2$, which is purified using a series of condensation traps before
being led into an isotope-ratio mass spectrometer (MAT253, by ThermoFinnigan) through a dual inlet
interface. Stable isotope ratios were calculated from the ratios of $CO_2$ masses 44, 45 and 46 analysed
by the mass spectrometer and converted to delta values with reference to the international Vienna Pee
Dee Belemnite (‰VPDB) scale using the standards NBS19, NBS18, CO-8 and LSVEC (IAEA, Vienna,
Austria). Stable isotope results were determined to be reproducible with a standard deviation less than
0.2 ‰ based on repeated measurements of Carrara marble powder. To compare our enamel-bound
stable oxygen isotope results with modern and Miocene water oxygen isotope values obtained in
previous studies, we used the enamel-water conversion formula for large, water-dependent herbivores
from (Hoppe, 2006) and we aligned the VSMOW to VPDB scale using the formula in (Coplen et al.,
1983), following the methodology in (de Rooij et al., 2022) and (Wooller et al., 2021).

### 4. Results and interpretation

4.1 Mineralogical characterization of sediments associated with studied molars
The X-ray diffraction analyses on Mid-Miocene soil samples revealed quartz as the main constituent
(*ca.* 95%; Table 1). Samples extracted from areas surrounding the teeth (inside and around) show lower
abundance of quartz (absent or 21 to 67%; Table 1) accompanied by a significant contribution of iron-
rich minerals (goethite, siderite, ilmenite). In contrast, modern sediment samples representing the host
substrate of the Lisbon Zoo are composed of approximately equal amounts of quartz and calcite (52



and 48%, respectively; Table 1). Calcite is not significantly present at any sample retrieved from Mid-
Miocene materials. Mid-Miocene sediment samples correspond to detrital quartz-grain deposits,
cemented into hard iron-rich crusts encasing the studied molars. Such "induration" is common in semi-
arid regions because of epidiagenetic processes (*sensu* Staunton and Fairbridge, 2008), typically found
in flood plain alluvium, lacustrine and littoral deposits affected by strong seasonality: wet season leads
to dissolution and the dry season to capillary action and evaporation (Beauvais and Colin, 1993;
Staunton and Fairbridge, 2008). The presence of dolomite and/or gypsum (see Table 1) is also
consistent with strong evaporation scenario (Díaz-Hernández et al., 2013). In contrast, the sediments
associated with the modern elephant molar show no evidence of such evaporation processes.
Clay mineral fractions also provided clear differences between Mid-Miocene and modern samples (Fig.
3): In ancient samples only illite was detected, whilst the modern sample is also dominated by illite
(84%) but also contains a small amount of smectite (16%; Fig. 6). Nevertheless, these differences may
be considered as negligible in terms of original composition of the ancient clay assemblage for two
reasons: (i) Smectite decays readily under shallow burial conditions (illitization of smectite) as water is
expelled from the interlayer space and sheets become more and more organized, leading to a well
crystallized illite crystal structure (Kübler and Jaboyedoff, 2000). It is thus plausible that smectite was a
part of the initial composition of the clay assemblage of mid-Miocene soils, as seen for the modern soil
sample (Fig. 6). (ii) Sediments reflecting a clay mineral composition of 100% illite are highly uncommon
(Journet et al., 2014), corroborating the probable illitization of original smectite. Therefore, the original
clay mineral assemblage at both sites can be considered similar, but further comparisons are limited,
as soil substrate at the Lisbon Zoo may not necessarily reflect local sediment composition (as described
in section 2.2).

4.2. Stable carbon and oxygen isotope composition of tooth enamel bioapatite
Mid-Miocene molar samples present $\delta^{13}$C values between -9.2 and -9.9‰VPDB (mean value ± standard
deviation (s.d.) = -9.7 ± 0.3‰VPDB; Fig. 7), in clear contrast with the modern elephant molars, whose
values range from -11.9 to -13.2‰VPDB (mean ± s.d.=-12.8 ± 0.3‰VPDB). Differences in $\delta^{18}$O are also
clear, with Mid-Miocene samples clustering between -1.7 and -6.7‰VPDB (mean ± s.d.=-3.5 ±
1.0‰VPDB) and modern samples between -7.9 and -11.1‰VPDB (mean ± s.d.=-8.6 ± 0.7‰VPDB). The
extent of diagenetic alteration can be evaluated by assessing the degree of isotopic heterogeneity
and/or homogeneity among specimens from a single deposit(Kohn and Cerling, 2002). In this way, the
observed clustering of all Mid-Miocene samples against the tight cluster formed by the modern molar
samples is taken as evidence of fair preservation of the older molars. This interpretation is further
validated by contrast with available literature for coeval materials and (paleo) environmental
conditions deduced for the factors controlling the obtained isotopic differences. When compared with
previous studies, we find that both our modern and our *Gomphotherium* $\delta^{13}$C data aligns well with
previous $\delta^{13}$C measurements for these taxa. The $\delta^{18}$O values measured in *Gomphotherium* for this
study are high compared to most literature $\delta^{18}$O values of the same genus, while our modern elephant
$\delta^{18}$O values are slightly lower than those of other modern and Pleistocene elephants (Figure 7).

4.3 Elemental composition of molar enamel
Semi-quantitative XRF maps highlight the distribution of multiple elements throughout the tooth
specimens (see elemental maps in https://doi.org/10.5281/zenodo.14882824). The maps show clear
features elevated in the concentrations of elements such as Fe (example in Fig. 5), which indicate patina
or encrustation on the outside of the teeth or cracks in the teeth which are enriched in these detrital
elements. The cementum cavity and dentine are also enriched in these elements compared to the
enamel layers. These areas were avoided as much as possible during line scanning to exclude the
influence of these postmortem processes from our interpretation. Outside these clearly diagenetic
features, the enamel of the *G. angustidens* specimens studied here is broadly similar to that of the



modern elephant, and to that of previous other modern mammal tooth enamel (De Winter et al., 2016;
Kohn, 2008) attesting to its preservation.
A summary of mean elemental values obtained for selected elements (Ca, P, Na, Mg, Sr, S, Si, Fe, Al, K
and Mn) based on quantitative XRF line scans is shown in Table 2 and Figure 8. Because of a wide array
of expected differences amongst the studied molars (geological age, geographical origin, taxonomic
difference), overall discrepancies in absolute values are beyond the scope of this contribution. Yet, due
to the potential interaction with surrounding soil and diagenetic incorporation of Si, Fe, Al, K (e.g.,
Białas et al., 2021; Brügmann et al., 2012), the fact that these elements show no clear enrichment
trend when comparing modern with ancient values seems to point towards a fair preservation of the
elemental composition of Mid-Miocene molars (see Table 2).

Iron is the only exception, but likely because Miocene molars are encased on iron-rich crusts (section
4.1). If iron oxide encrustation caused elevated Fe in the fossil specimen, such enrichment should be
higher on the outside of the molars, a pattern visible in the line scan results (see below) and the XRF
maps (see Appendix; https://doi.org/10.5281/zenodo.14882824). In this case, elemental fluctuations
observed in phase with Fe are suspect of diagenetic influence and not interpreted in the discussion. In
fact, due to eliminate risk of discussing differences in absolute elemental values due to diagenetic
alteration, including depletion or enrichment of some elements (Fig. 8 and Table 2), we focus on
discussing elemental trends during tooth growth and elemental associations, as these features may
enclose (paleo) environmental fluctuations during the growing period (typically <4/5years (Uno et al.,
400  2020).


Elemental associations were obtained by performing Principal Component Analysis (PCA) on Mid-
Miocene and modern elemental data (Fig. 9). Details on preliminary data treatment and exploratory
analysis are provided in Supplementary File (Figs. S1 to S4). Principle Component Analysis has been
shown to help detecting diagenetic signatures in fossil archives, because elements associated with
diagenetic alteration (e.g. Mn and Fe in the case of carbonates (Ullmann and Korte, 2015) or rare earth
elements, U and Mn and Fe in the case of bioapatite (de Winter et al., 2019; Kohn, 2008; McMillan et
al., 2019) tend to cluster into a clearly distinguishable principle axis due to their strong enrichment in
altered parts of the fossils (Coimbra et al., 2020). Modern and ancient samples provided very similar
PCA loading plots (Fig. 9A and B). Principle Component 1 (PC1) is characterized by an opposing trend
of Ca and P concentrations on the one hand against concentrations of Fe, Si, Al, K (with minor
contributions of Mn and Sr) on the other. Principle Component 2 (PC2) represents a trend of Ca and P
concentrations contrasted to samples with higher Na, Mg and Sr concentrations. Accordingly, lowered
Ca and P values are associated to increased abundance of the remaining elements ("closed sum effect";
Fig. 9A and B), but without relation between the groups of elements separated in different components
(PC1: Fe, Si, Al, K; PC2: Na, Mg, Sr). Sample distribution along the PCA score plots (Fig. 9C and D) shows
a main cluster from which a small number of samples deviate towards higher concentration of Fe, Si,
Al, K (negative PC1 values; horizontal axis) and higher concentration of Na, Mg, Sr (positive PC2 values;
vertical axis). Because these elemental associations are seen in both modern and ancient samples,
diagenetic control on obtained elemental distributions is unlikely (also shown during exploratory data
analysis; Supplementary File). The observed cluster distribution suggests that incorporation of Fe, Si,
Al, K and Mn occurs only sporadically and may partly highlight incorporation of small fractions of
detrital matter on the edges of the line scans where they exit the enamel. In contrast, Na, Mg and Sr
incorporation is recurrent throughout the elemental dataset, suggesting consistent variability within
the enamel which may be linked to *in vivo* environmental variability recorded in the teeth (see (De
Winter et al., 2016).





The variability in concentrations of the statistically most relevant elements from each cluster
defined by PCA (Ca, Na, Fe and Si) along molar growth was tested by representing the fluctuations in
their abundance along the measured transects (Fig. 10). Along transects in growth direction through
modern elephant molars (Fig. 10A), Ca and Na show pronounced fluctuations, which are inversely
correlated. This relationship between Ca and Na is only rarely disrupted when Fe and Si concentrations
are conspicuously higher, in which case Ca and Na both decrease (Figure 10. The mid-Miocene
elemental transects reflect the same trend (Fig. 10B) with an anti-phase pattern of Ca and Na with
significant fluctuations within one single transect. Concentrations of Ca and Na jointly decrease in
locations where Fe and Si show sharp increases. The evolution of Fe and Na along the transects, which
sample the same part of the ontogenetic history in each molar were further compared to evaluate their
potential as seasonality archives (Figs. 11, 12 and 13). The resulting comparison shows that both
episodic increases in Fe concentration and periodic fluctuations in Ca and Na concentrations in the
modern elephant and Miocene *G. angustidens* molars are reproduced along parallel transects in the
same tooth.

**5. Discussion**

**5.1 Stable isotope signatures in bioapatite**
5.1.1    Feeding preferences derived from $\delta^{13}$C signature in Mid-Miocene and modern molars
The stable carbon isotope composition ($\delta^{13}$C value) of body tissue of extant herbivores reflects
the type of vegetation ($C_3$ vs. $C_4$) they consume. Terrestrial plants may follow three types of metabolic
pathways to fix $CO_2$ from the atmosphere: the $C_4$, $C_3$, CAM pathways (Bender, 1971; Ehleringer et al.,
1991; Farquhar et al., 1989). The $C_3$ and $C_4$ photosynthetic pathways exhibit different $\delta^{13}$C values, with
most $C_4$ plants (tropical grasses and sedges) containing carbon with an isotopic value between -10‰
and -15‰VPDB and most $C_3$ plants (most trees, shrubs, or high-latitude grasses) between -22‰ and -
35‰VPDB (Bender, 1971; Vogel, 1980). The distinct isotopic composition of $C_3$ and $C_4$ plants translates
to a distinct isotopic signature in tooth enamel carbonate that can be used as a proxy for browsing
(mostly $C_3$) versus grazing (mostly $C_4$) diet of animals. The typical range of $\delta^{13}$C values for these different
feeding strategies shown in Fig. 7 to compare with our results were obtained from (Cerling et al., 2003b,
2003a; Cerling and Harris, 1999; Clementz, 2012; Martínez del Rio and Carleton, 2012). Metabolic
pathways in an animal increase the $\delta^{13}$C values by up to 14‰ in the dental enamel apatite relative to
plant tissue they consume (Cerling and Harris, 1999; Patnaik et al., 2019). Further, animals feeding in
closed canopy forests have a lowered $\delta^{13}$C composition compared to those feeding in more open
woodland environments due to the recycling of respired $CO_2$ ($^{13}$C depleted) on the forest floor and low
light intensities at ground level that results in more negative $\delta^{13}$C values in plants (Cerling et al., 2004;
Patnaik et al., 2019; van der Merwe and Medina, 1991). For the present study, additional aspects are
of relevance. Indeed, $\delta^{13}$C values in modern elephant molars are expected to be *ca.* 1.5‰ lower relative
to Miocene *G. angustidens* values due to a shift in atmospheric $\delta^{13}$C since the Industrial Revolution and
fossil fuel combustion (Cerling et al., 1997b, p. 97; Passey and Cerling, 2002; Tipple et al., 2010).
The range of $\delta^{13}$C values obtained for Mid-Miocene samples largely overlaps the range
commonly reported for Mid to late-Miocene *Gomphotherium* samples from a variety of locations (Fox
and Fisher, 2004; Patnaik et al., 2019; Wu et al., 2018); Fig. 7). All reported values plot along the upper
extreme range for $C_3$ consumers (-8 to -12‰VPDB; Fig. 7), indicating that—also in Iberia—these
individuals foraged in partially open, possibly arid conditions, favouring woodlands as their preferred
habitat. Additionally, lack of evidence pointing towards of mixed-feeding preferences indicates that
western Iberian areas of woodland habitat could support these large-bodied herbivores through the
Mid-Miocene. In clear contrast, modern elephant $\delta^{13}$C values obtained in this study as well as from
other localities (Ma et al., 2019; Uno et al., 2020); Fig. 7) plot towards lower values (-10 to below -
14‰VPDB), within the range of closed canopy forest environments. Regarding the modern samples



used in this study, the obtained result suggests that the animal developed the molar in the wild, prior
to transport to its zoo environment.

5.1.2    Hydrological differences between Mid-Miocene and present-day environments
The $\delta^{18}O$ of tooth enamel reflects the $\delta^{18}O$ of ingested water, largely influenced by precipitation,
latitude, altitude, aridity, and evaporative processes, as well as physiological/behavioural water
conservation factors and metabolic processes of mammals (Kohn, 1996; Kohn et al., 1996, p. 96; Luz et
al., 1984). Obligate drinkers, such as elephants, frequently ingest water, which largely reflect rainfall
oxygen-isotope composition whilst more drought-tolerant non-obligate drinking mammals (e.g. *Oryx*)
are affected by evaporative enrichment of $\delta^{18}O$ values (Ayliffe and Chivas, 1990; Kohn et al., 1996; Levin
et al., 2006; Roberts et al., 2018). Concomitantly, environmental temperature changes (e.g., seasonal)
lead to the enrichment of the $\delta^{18}O$ signature in the water source in warmer conditions and relative
depletion in cooler conditions (Bryant et al., 1996).
The range of $\delta^{18}O$ values obtained for Mid-Miocene molar samples is similar to the high end of the
range of $\delta^{18}O$ values measured in North American Mid to Late-Miocene *Gomphotherium* molar samples
found in the literature ((Fox and Fisher, 2004); Fig. 7), but significantly higher (less depleted) than values
reported for Indian and Asian Mid-Miocene *Gomphotherium* molars ((Białas et al., 2021; Patnaik et al.,
2019; Wu et al., 2018); Fig. 7). Investigating the expected $\delta^{18}O$ values of precipitation along these
different areas of the globe during the Miocene provides clues to explain these differences. The
obtained oxygen isotope values in the Mid-Miocene molar samples overlap almost exactly with $\delta^{18}O$
values modelled for rainfall during dry season in western Iberia during the Middle Miocene (Botsyun
et al., 2022). Similarly, oxygen isotope compositions in Asian *Gomphotherium* molars are also
compatible with pedogenic carbonate $\delta^{18}O$ values from the Chinese Loess Plateau and northern
Pakistan, which record the isotopic value of local surface waters in these Asian region (Quade et al.,
1989; Wang et al., 2023, p. 23). Considering this relationship, the comparison between $\delta^{18}O$ values
recorded in modern and fossil molars reveals that Mid-Miocene weather in western Iberia was likely
warmer and/or less humid than in Asia during coeval times.
Comparing modern elephant enamel $\delta^{18}O$ values and local precipitation values in SW Iberia generates
a significant discrepancy of ca. 5‰ (Fig. 7). Such a difference is not observed in other modern samples
(Fig. 7), which show a very consistent overlap in $\delta^{18}O$ values between $\delta^{18}O_{enamel}$ and $\delta^{18}O_{precipitation}$.
Therefore, as already suspected from the interpretation of $\delta^{13}C$ values, the modern elephant living at
the Lisbon Zoo is unlikely to have developed his molar in that locality, thus reflecting $\delta^{18}O$ values of the
original location in the wild. Based on considerably depleted $\delta^{18}O$ values (ca. -8.5‰), possible origins
of the modern specimen include central Angola, southern Democratic Republic of the Congo or Zambia,
where the more depleted range of surface water $\delta^{18}O$ values overlaps with $\delta^{18}O$ values of drinking
water expected based on the enamel-$\delta^{18}O$ values in our modern elephant specimen ("Water Isotopes
Database," 2023).
In summary, carbon and oxygen-isotope values obtained from Mid-Miocene molar samples
demonstrate an open-canopy woodland habitat preference for West Iberian *Gomphotherium*. Local
food supply sustained browsing communities even during dry season, as no evidence of grazing was
detected in $\delta^{13}C$ values. Therefore, warm and dry conditions seem to have prevailed in Western Iberia
during the Mid-Miocene. To isolate seasonal-scale variability in the paleoenvironment, high-resolution
elemental data are gathered and interpreted below.

**5.2 Elemental associations and chemical variability**
5.2.1    Trace element concentrations in tooth enamel



Tooth enamel is regarded as one of the most diagenetically inert biominerals and its chemical and
isotopic compositions are preferred paleoenvironmental and paleoclimatic proxies (Forbes et al., 2010;
Fricke and O'Neil, 1996; Kohn, 1996; Kohn and Cerling, 2002). During precipitation, enamel biomineral
composition not only respond to the local chemical environment, i.e. the bioavailability of chemical
elements, but also to physiological and taxonomical characteristics. Therefore, our research design
comparing related proboscidean taxa aims to isolate climatic and environmental information from the
chemical composition of the bioapatite samples.
The nano- to microcrystalline structure of tooth bioapatite has several sites for cations and anions,
which permits the uptake of a variety of elements with rather different chemical features. These
elements find their way from the environment into plant and animal tissue, including bone and tooth
material. For this reason, the chemical and isotopic composition of bioapatite became an important
tool to monitor climatic and ecological change (Kohn and Cerling, 2002; Macfadden et al., 1999; Tütken
et al., 2007). The bioapatite mineral lattice can accommodate iso- and heterovalent substitutions
during life or diagenesis varying its chemical composition through (geological) time, making bioapatite
a unique archive of physical and chemical information for both the living cycle and the events occurring
after death (Malferrari et al., 2019).

5.2.2 Differential incorporation of major and trace elements in molar enamel
The inorganic component of mineralized enamel is composed of 89% calcium hydroxyapatite
and small amounts of calcium carbonate, calcium fluoride and magnesium phosphate. Biological
apatite in enamel (and dentin) are deficient in calcium, rich in carbonates and prone to structural
element replacement (LeGeros, 1986). Ionic substitutions of enamel apatite (bioapatite) include $Mg^{2+}$,
$Na^+$, $Mn^{2+}$, $Ni^{2+}$, $Cu^{2+}$ or $Zn^{2+}$ for $Ca^{2+}$; $Cl^-$ or $F^-/CO_3^{2-}$ for $OH^-$; and $HPO_4^{2-}$, $CO_3^{2-}$, $AsO_4^{3-}$, $SO_4^{2-}$ for $PO_4^{3-}$
(Sakae et al., 1991; Sarna-Boś et al., 2022; Shaik et al., 2021). During lifetime, demineralization and
remineralization processes are constantly alternating on the surface of each tooth, also contributing to
potential changes in elemental composition of teeth (Nedoklan et al., 2021).
Magnesium is the most abundant Ca-substituting trace element in enamel, essential for the
proper development of the tooth structure (Sarna-Boś et al., 2022; Sharma et al., 2021). Magnesium
(and sodium) can replace Ca by ion exchange on the crystal surface during the secretion and maturation
of primary bioapatite. It may occur in a labile status, probably adsorbed onto the crystal surfaces (Aoba
et al., 1992a, 1992b). The elemental association obtained for both modern and ancient proboscidean
molars, which clearly show an inverse trend between Ca and P concentrations when compared to
abundances of Mg, Na and Sr (Fig. 9A and B), indicates that Ca replacement within enamel crystal
surfaces occurred naturally during molar development and/or during remineralization processes during
the animal's lifetime. The fact that the largest cluster of samples is responding to this trend (Fig. 9C and
D) shows that this process is dominant over time, corresponding to tooth growth and later
maintenance stages, and not principally related to post-mortem diagenetic processes.
As observed by other authors, small amounts of externally incorporated elements (Si, Al, and
Fe, Mn and rare-earth elements) can also be found in fossil enamel, as long-term preservation of
bioapatite involves recrystallization and alteration processes, which drive enrichment of elements not
naturally abundant in unaltered enamel (Malferrari et al., 2019). Accordingly, minor enrichment or
depletion of key elements near the outer enamel rim may indicate the mobilization of these elements
into the bioapatite structure during diagenesis (see Fig. 5 for Fe; (de Winter et al., 2019; Kohn, 2008).
Conversely, lack of significant elemental changes in enamel Fe (and Mn) from the outer rim of the tooth
towards the innermost sections, as seen in our *Gomphotherium* specimens (see Fig. 11), denotes
inefficient impregnation of interstitial Fe- and Mn-oxides and oxyhydroxides (Kohn et al., 1999).
Additionally, lack of a clear difference in the elemental distribution between modern to fossil data is
typically taken as a good indicator that diagenesis had no significant influence on their distribution
(Białas et al., 2021). In conclusion, the following previously established lines of evidence for screening



for diagenetic influence on elemental enamel composition led us to conclude that our *G. angustidens*
dataset reliably records past dietary and environmental change:
1.   Modern and ancient samples show similar elemental associations (Fig. 9A and B) and age-
dependent elemental enrichment in Fe, Si and Al is not observed (Table 2);
2.   There is no clear pattern of trace element enrichment along the outer rims affecting the
locations of the selected transects (Fig. 10);
3.   Only a small set of samples (13 and 17% of modern and ancient data; respectively) shows
elevated abundance of key elements associated with diagenetic alteration (as isolated in PC1;
Fig. 9C and D), showing this to be a sporadic enrichment events, rather than a pervasive
process.

Even at early diagenetic stages, local sedimentary factors influencing the mobilization of FeO,
F and $SO_3$, such as rock composition, pH value and redox potential, can control the introduction of
these elements into the tooth materials during fossilization. Similarly, comparable concentrations of
$K_2O$, $SiO_2$, and $Al_2O_3$ in both modern and fossil materials indicate that crystallization or mechanical
introduction of silicate phases into the fossil during diagenesis is of low importance. Therefore, even
elements typically attributed to early diagenetic evolution (Fe, Si, Al, K) can be interpreted to reflect *in*
*vivo* processes similar to the distribution of other major to minor elements (e.g., Brügmann et al.,
2012). Short-lived periods of significant incorporation of Fe, Si, Al, K are inversely correlated to tooth
growth and maintenance processes evidenced by reduced Mg and Na fixation (Figs. 9 and 10). This is
taken as evidence of periods of environmental stress.

5.2.2. Elemental abundance in *Gomphotherium* molars reveals geophagic behaviour in ancient
proboscideans

The patterns of trace element concentration identified in the XRF line scans through the
*Gomphotherium* tooth enamel record differential elemental incorporation during the animal's lifetime.
Potential sources of elemental enrichment and causes leading to elemental depletion must be
interpreted in the light of natural behaviours carried out in response to environmental conditions
(Wheelock, 1980; Białas et al., 2021). Amongst these, geophagy is commonly observed in elephants.
Geophagy is defined as the deliberate and regular consumption of earthy materials such as soils, clays
or sediments by animals and humans (Abrahams and Parsons, 1996) and functions to supplement
dietary mineral deficiencies, alleviate gastrointestinal disorders or detoxify unpalatable foods
(Chandrajith et al., 2009; Holdø et al., 2002; Houston et al., 2001; Sach et al., 2019; Wheelock, 1980).
Soils consumed by African elephants were previously demonstrated to have higher $Na^+$ concentrations
than the surrounding soils, hence deduced to be used to supplement an inadequate dietary-sodium
intake (Chandrajith et al., 2009; Wheelock, 1980). However, other possible roles of soil have also been
hypothesized as important in determining the extent of geophagy, including detoxification of plant
secondary compounds, countering the effects of acidosis (Houston et al., 2001). Mineral
supplementation is most likely not the sole factor driving geophagic behaviour (Holdø et al., 2002).

Further evidence for the relevance of geophagy and local sedimentary contexts in modern
elephants is deduced from the fact that soil used by elephants at Ngorongoro (North Tanzania) contains
ca. 35% kaolinite (Houston et al., 2001). Kaolinite, through its adsorptive ability, is effective in
neutralizing the activity of many plant secondary (toxic) compounds also serving as a barrier to protect
the gut lining from toxins in the food (e.g., tannins and alkaloids; Sach et al., 2019). In this way, forest
elephants, which have access to kaolin soils, may be able to feed on a wider range of forest plant
species, which can be beneficial under nutritional stress resulting from low food quality during dry
season (Suba et al., 2016). Elephants purposefully select soils with high proportions of kaolinite and
illite, particularly during the dry season when food quality is relatively low (Chandrajith et al., 2009). In
this context, the clay assemblage recovered from both modern and ancient materials (Fig. 6),
dominated by illite [$KAl_2(Si_3, Al)O_{10}(OH)_2$] and small contribution of smectite [$(Na,Ca)_{0,3}(Al,Fe,Mg)_2(Si,$



Al)$_4$O$_{10}$(OH)$_2$•n(H$_2$O)] explains the incorporation of minor elements via geophagic behaviour, providing
a viable source of Fe, Si, K, Al (Figs. 9 and 10). Illite is reported to be the most abundant clay mineral in
land surfaces at a global scale (Ito and Wagai, 2017).
An alternative hypothesis explaining these spikes in Fe, Si and Al in our trace element profiles
could be the incorporation (either *in vivo* or postmortem) of fine-grained sediment into cracks in the
enamel. This would not require digestion of the minerals, but, if incorporated *in vivo*, would still point
to geophagic behaviour. After closely examining the distribution of these elements in XRF maps (Figure
10 and Appendix; https://doi.org/10.5281/zenodo.14882824) as well as the texture of the enamel at
the location of the XRF line scans, we consider this explanation unlikely, as no evidence of cracks in the
enamel is found at the measurement locations. This is a result of our measurement strategy, in which
we deliberately avoided these cracked areas when programming our XRF profiles (see **section 3.3.2**).
We hypothesize that the short-lived, sporadic increments in Fe, Si, Al in both modern and
ancient molars result from elemental incorporation due to geophagic behaviour during stressful, dry
season conditions. By comparing Fe trends along all measured transects (Fig. 11), it is possible to
identify the occurrence of several dry periods during molar growth (Fig. 11B, D, F and H). It is not
possible to confirm their seasonal periodicity with certainty here, as enamel growth rates cannot be
independently reconstructed, limiting the estimation of the amount of time comprised in each
transect. Tentatively, three to four dry seasons are identified based on the sharp Fe increments
coinciding with cyclical patterns in Na, suggesting that the obtained elemental archive recovers a period
of 3 to 4 years in the life of these animals. Based on the reasoning above, even small amounts of
externally incorporated elements (Fe, Si, Al) can serve as proxies to identify seasonality in modern and
ancient molar samples if the animals participated in seasonal geophagy. This result shows that Miocene
seasonality was sufficiently pronounced to have an impact on animal behaviour, stimulating geophagic
behaviour under seasonally stressful (warm and dry) conditions. Complementing these findings, Na
distribution likely indicates variations in tooth growth, with higher Na values highlighting growth under
optimal conditions (Fig. 12). The Na record showing smooth long-duration increments along one to
three periods (Fig. 12B, D, F and H) supports the interpretation that Na in our *Gomphotherium* can be
used as a proxy for seasonal changes in tooth growth rate, while the Fe spikes highlight periods of
drought, geophagy and reduced growth. This observation is supported by evidence of Na deficiency as
one of the drivers of geophagic behaviour and natural preference for Na-rich sediments (Chandrajith
et al., 2009; Wheelock, 1980). The timing of elemental signatures in the trace element records, which
reflect dry season, does not overlap with those interpreted to reflect more optimal growth conditions
(Fig. 13), which corroborates the usefulness of differential elemental incorporation as indicators of
seasonal variability in environmental conditions in ancient teeth. Based on these findings, the
applicability of trace elemental concentrations in well-preserved fossil enamel bioapatite as proxies
for geophagy behaviour and growth stress during past dry seasons is encouraging.

**6. Conclusions**
A combination of stable isotope and trace element analyses in molar enamel in Middle Miocene *G.*
*angustidens* from Western Iberia and modern *L. africana* from western Iberia reveals similar responses
of tooth bioapatite chemistry to episodic changes in the environment in both taxa. Based on stable
carbon and oxygen isotope composition in both taxa, Iberia was characterized by a warm and
seasonally dry climate during the Middle Miocene, and that *G. angustidens* thrived in an open-canopy
woodland ecosystem in this area. In contrast, the modern *L. africana* specimen developed his molar in
a Central African closed-canopy forest environment. Episodic increases in Fe and Si observed in trace
element profiles through *G. angustidens* molar enamel hint at geophagy behaviour during seasonally
dry and warm periods in this extinct taxon, behaviour patterns, which are also observed in modern *L.*
*africana*. Cross-validation via independent proxies suggests surrounding sediment as a probable source
of elemental enrichment, incorporated during molar development and/or during remineralization



processes. These results demonstrate the application of high-resolution (sub-seasonal scale) trace element records through fossil teeth is critical to reconstruct the behaviour patterns of extinct mammals in response to climate variability in their living environment.

**7. Acknowledgments**

RC acknowledges financial support from Geobiotec research Group (UIDB/04035/2020). MR thanks the Stimulus of Scientific Employment, Individual Support – 2018 Call grant by the Fundação para a Ciência e a Tecnologia (Portugal, CEECIND/02199/2018) and GeoBioTec as well as the Fundação para a Ciência e a Tecnologia for the funding of the project 2021        EXPL/CTA-PAL/0832/2021    which    directly supports this research. DEL is recipient of a PhD grant funded by Fundação para a Ciência e a Tecnologia (grant number 2020.05395.BD). RM is founded by Fundação para a Ciência e a Tecnologia (grant number 2021.08458.BD). We also thank Micael Martinho and Carla Alexandra Tomás from the preparation laboratory at GEAL-ML as well as Lígia Castro and Eduarda Ferreira from the Earth Sciences Department at FCT-NOVA. Finally, a special thanks to the project manager Marlene Monte from NOVA.ID.FCT. PC thanks Research Foundation Flanders (FWO) for purchasing µXRF and VUB Strategic Research for funding. NdW acknowledges financial support from the Flemish Research Council (FWO postdoctoral fellowship; 12ZB220N and FWO Climate Prize) as well as the Dutch Research Council (NWO) for his VENI fellowship ("MACRO"; VI.Veni.222.354).

**Data availability.** Supplementary data and figures belonging to this study were stored in the open-source online repository Zenodo (https://doi.org/10.5281/zenodo.14882824)

**Author contributions.** RC, NdW, and MR conceived the work; all authors wrote, reviewed and edited the manuscript. Samples were collected by MR, RB, DEL, PL and RM; laboratory procedures were supervised and carried out by RC, NdW, MR and AG. FR and PC provided the framework, facilities and necessary support to develop this research.

**Competing interest.** *Some authors are members of the editorial board of journal Biogeosciences.*





**Figure captions**

**Figure 1.** Relevant aspects of Proboscidean morphology and dental anatomy. A) Artistic illustration of
*Gomphotherium angustidens* and scale relative to human silhouette (adapted from ("*Gomphotherium*
*angustidens* by cisiopurple on DeviantArt," 2022), ©cisiopurple). B) Schematic representation of molar
growth process (adapted from (Jayachandran, 2022) ©Wildlife SOS). C) Molar structure and relative
distribution of enamel, dentine and cement (adapted from ("What Did Lupe Eat? - Mammoth
Discovery," n.d.) ©Children's Discovery Museum, San Jose, USA). E) Lophodont of elephant in surface
view, describing the intricate folding of enamel and dentine to form transverse ridges or lophs (adapted
from Bhavya, 2017).

**Figure 2.** Geographical setting of the studied areas. A) Location of the Portuguese mainland in the
Iberia Peninsula. B) Regional distribution of main geological outcrops at the Lisbon area (adapted from
LNEG-LGM, 2010). Squares indicate the location of studied sections. C) General biochronostratigraphic
scheme for the Miocene and stratigraphic context of studied deposits at the Lisbon area (adapted from
Pais et al., 2012). D and E) Field aspect at the localities where the samples were extracted. The molars
analysed in this study originate from the Quinta da Farinheira in the Lower Tagus Basin (Areias do Vale
de Chelas, Lisbon). The Chelas unit has been designated SDL1 by Antunes & Ginsburg (2003) and Vb by
Cotter (1956), with an estimated age of (~15.9–16.1 Ma) and is highlighted by a red arrow. It belongs
to the Langhian stage and the MN5 biozone (13.7–16.0 Ma).

**Figure 3.** General aspect of uncut molar samples. A to C) *Gomphotherium angustidens* molars, note
less significant wear of the collected samples from A to C. D) Modern L. Africana molar, showing typical
diamond-shaped ridges.

**Figure 4.** Microscope composite of **l**ongitudinally cut modern and Mid-Miocene molar samples with
indication of elemental transects chosen for each section (red arrows). Note the inwards-outwards
direction of all transects, along growth direction Specimen FCT-DCT 4945 in the top right, FCT-DCT
4933 bottom center and FCT-DCT 4943 on the far right. The modern elephant molar is depicted in the
bottom left.

**Figure 5.** Relative elemental abundance for all specimen, here shown for iron and strontium
(distribution of other elements can be found at https://doi.org/10.5281/zenodo.14882824). Note the
lack of areas denoting higher iron abundance along selected transects and the clear Sr concentration
pattern highlighting well-preserved tooth morphology.

**Figure 6.** Clay mineral assemblage for Mid-Miocene sediment samples collected from areas adjacent
to the molars, around and inside cavities. Modern sediment sample reflects soil substrate from the
Lisbon Zoo (see text for details). Note the dominance of illite in all cases, along with minor contribution
of smectite for the modern sample.

**Figure 7.** Biplot representation of stable carbon and oxygen isotope values, highlighting the differences
between the two sets of samples tested (modern *vs.* Mid-Miocene). Shaded fields represent C and O
isotope values in enamel-bound carbonate reported by other authors (see reference list) for Mid-



Miocene and modern settings. Vertical arrows indicate range of variation of oxygen isotope values
representative of rainfall composition at different localities (modern and ancient). Oxygen isotope
values of water (right vertical scale) are compared to oxygen isotope values in bioapatite using the
formula for large, water-dependent herbivores from (Hoppe, 2006) and we aligned the VSMOW to
VPDB scale using the formula in (Coplen et al., 1983) following the methodology in (de Rooij et al.,
2022) and (Wooller et al., 2021) . Forest/woodlands range of carbon-isotope variation after Patnaik et
al. (2019) and references therein.

**Figure 8.** Boxplot representation (median and interquartile distance) of absolute elemental
concentrations (in ppm). A and B) note similar distribution of Ca and P; C and D) lowered values for
ancient samples regarding Na and Mg; E to K) overall higher median values for the remaining proxies
(except K). See Table 2 for full elemental range, mean and standard deviation.

**Figure 9.** Principal component analysis computed for the modern and Mid-Miocene samples. A and B)
Principal component loadings explaining 51 to 66% of the total variability of each set of samples. Note
similar distribution of elements along the PC axis. Modern Mn and S were not used due to low number
of measurements available (N<30%). C and D) Principal component scores indicating sample
distribution along the PCA space. Note main clusters of samples in both cases, with only few samples
(13% to 16% of the total number of samples) departing from the observed main trend.

**Figure 10.** Elemental trend of selected proxies (Ca, Na, Fe and Si) for all measured transects (see Figs.
4 and 5 for exact measurement sites). A) Trend lines for modern transects. B) Trend lines for Mid-
Miocene transects. Note overall similar records of Ca vs. Na and Fe vs. Si, as well as lack of clear
enrichment or depletion trends along each transect. Note opposite variation pattern for Ca and Na,
except when Fe and Si become significantly higher (white dashed boxes). Trend lines smoothed (10
points adjacent averaging); numerals according to Figure 4. Colors represent the different enamel
transects analyzed. The same color scheme is used consistently across all figures to maintain clarity and
allow for easy comparison of the different lines or transects.

**Figure 11.** Elemental trend line obtained for Fe measurements (line numbers as in Figure 4). A, C E and
G) Record of Fe abundance along growth direction (in mm). B, D F and H) Same datasets stacked and
slightly adjusted (vertically and horizontally) to compare different transects that represent similar
portions of the molar. (*) lines not to scale: mean horizontal scaling factor of 5.1; mean vertical scaling
factor of 3.3 (see Table S1 in Appendix). Drawing represents dry season periods. Colors represent the
different enamel transects analyzed. The same color scheme is used consistently across all figures to
maintain clarity and allow for easy comparison of the different lines or transects.

**Figure 12.** Elemental trend line obtained for Na measurements (line numbers as in Figure 3). A, C and
E) Record of Na abundance along growth direction (in mm). B, D and F) Same datasets stacked and
slightly adjusted (vertically and horizontally) to compare different transects that represent similar
portions of the molar. (*) lines not to scale: mean horizontal scaling factor of 5.1; mean vertical scaling
factor of 1 (see Table S1 in Appendix). Drawing symbol represents wet season periods. Colors represent
the different enamel transects analyzed. The same color scheme is used consistently across all figures
to maintain clarity and allow for easy comparison of the different lines or transects.



**Figure 13.** Compilation of elemental trend lines for Fe and Na as shown in Figures 11 and 12. A to C)
Mid-Miocene transects in growth direction showing alternating peaks of higher abundance of the
chosen elements. D) Modern transects in growth direction showing alternating peaks of higher
abundance of the chosen elements.

**Table 1.** Semiquantitative abundance (%) of bulk mineralogical composition of sediment samples.

**Table 2.** Mean (±standard deviation; s.d.), minimum and maximum elemental values obtained for
selected proxies obtained from Miocene and modern samples.

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

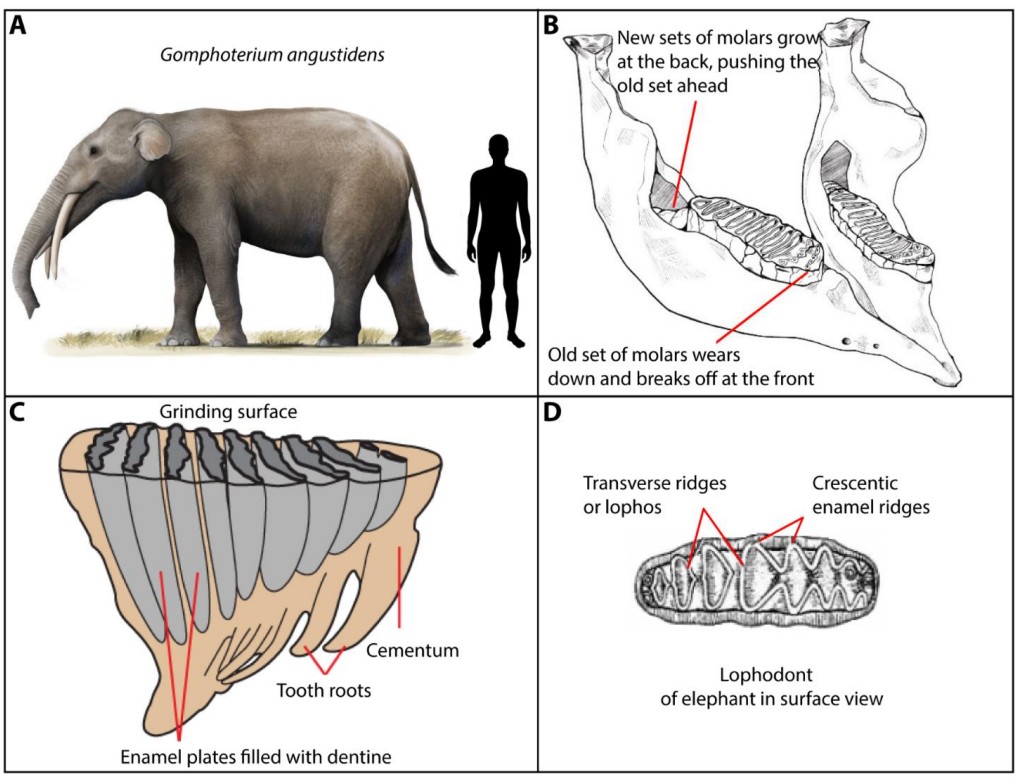

**Figure 1.** Relevant aspects of Proboscidean morphology and dental anatomy. A) Artistic illustration of *Gomphotherium angustidens* and scale relative to human silhouette (adapted from ("*Gomphotherium angustidens* by cisiopurple on DeviantArt," 2022), ©cisiopurple). B) Schematic representation of molar growth process (adapted from (Jayachandran, 2022) ©Wildlife SOS). C) Molar structure and relative distribution of enamel, dentine and cement (adapted from ("What Did Lupe Eat? - Mammoth Discovery," n.d.) ©Children's Discovery Museum, San Jose, USA). E) Lophodont of elephant in surface view, describing the intricate folding of enamel and dentine to form transverse ridges or lophs (adapted from Bhavya, 2017).



**Figure 2_Coimbra et al.**

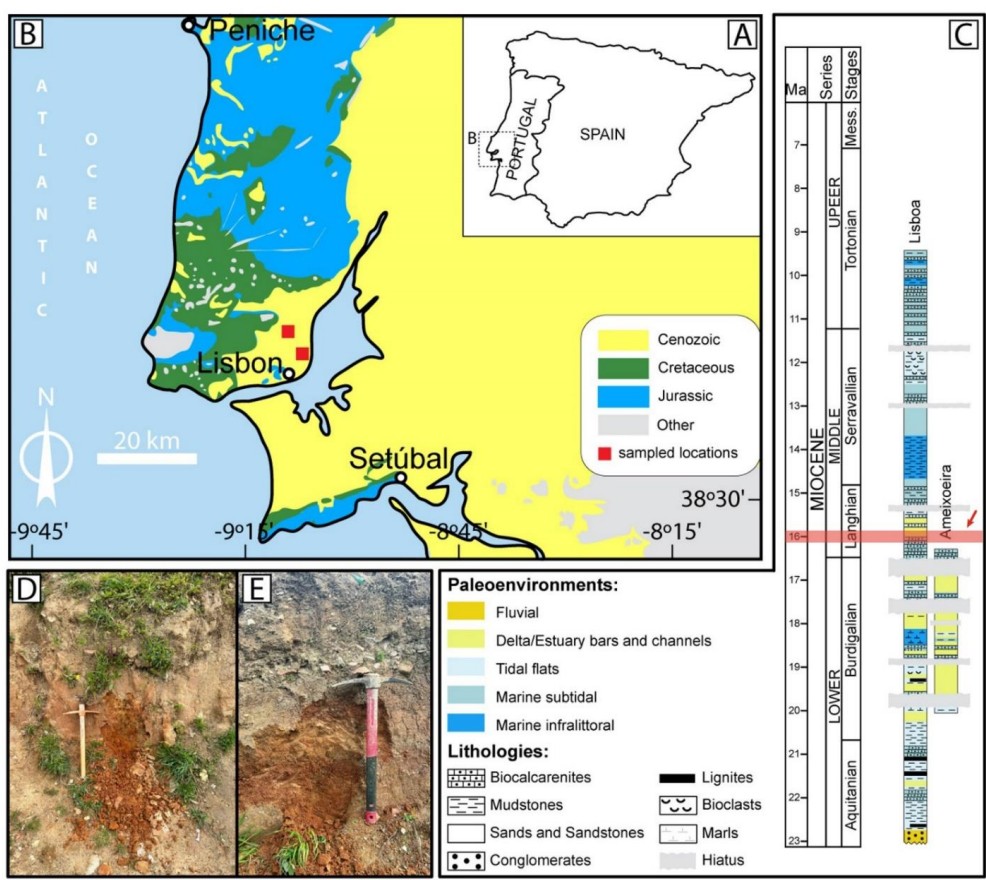

**Figure 2.** Geographical setting of the studied areas. A) Location of the Portuguese mainland in the Iberia Peninsula. B) Regional distribution of main geological outcrops at the Lisbon area (adapted from LNEG-LGM, 2010). Squares indicate the location of studied sections. C) General biochronostratigraphic scheme for the Miocene and stratigraphic context of studied deposits at the Lisbon area (adapted from Pais et al., 2012). D and E) Field aspect at the localities where the samples were extracted. The molars analysed in this study originate from the Quinta da Farinheira in the Lower Tagus Basin (Areias do Vale de Chelas, Lisbon). The Chelas unit has been designated SDL1 by Antunes & Ginsburg (2003) and Vb by Cotter (1956), with an estimated age of (~15.9–16.1 Ma) and is highlighted by a red arrow. It belongs to the Langhian stage and the MN5 biozone (13.7–16.0 Ma).



**Figure 3_Coimbra et al.**

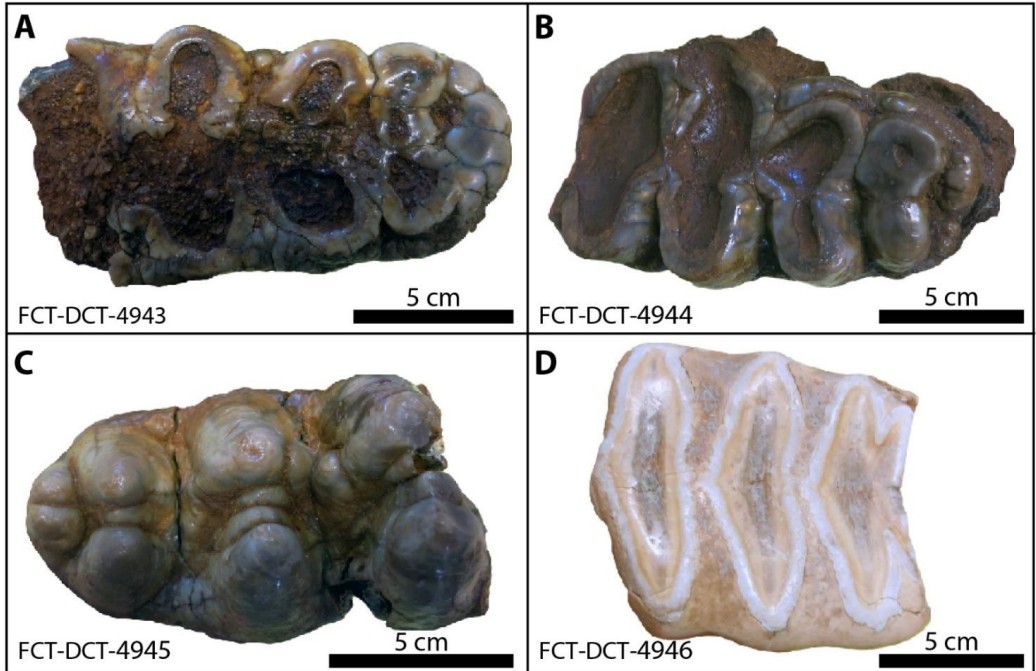

**Figure 3.** General aspect of uncut molar samples. A to C) *Gomphotherium angustidens* molars, note less significant wear of the collected samples from A to C. D) Modern L. Africana molar, showing typical diamond-shaped ridges.



**Figure 4_Coimbra et al.**

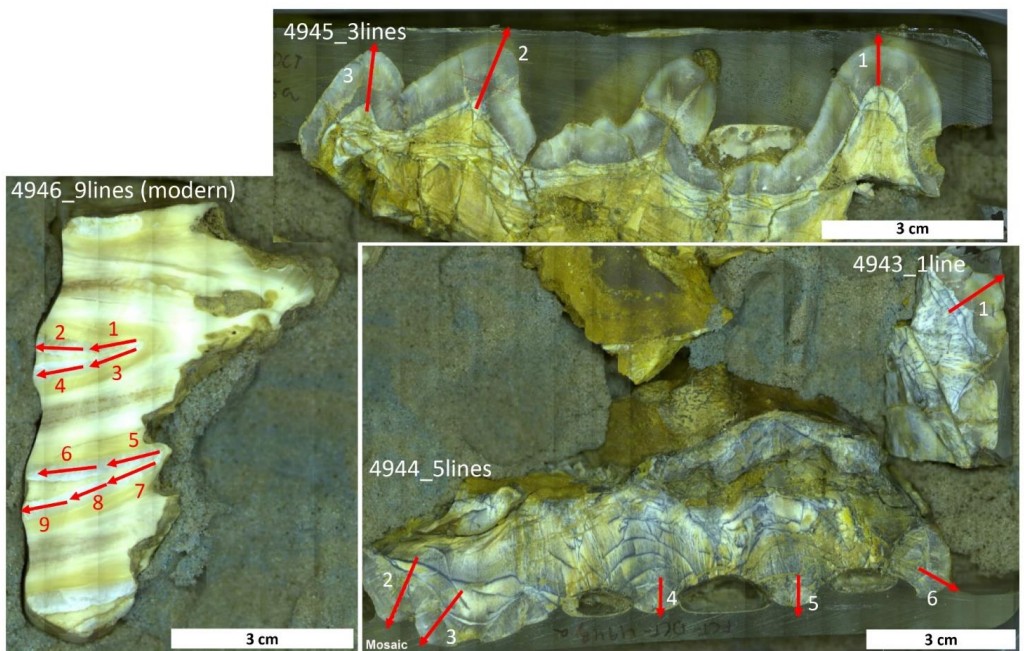

**Figure 4.** Microscope composite of longitudinally cut modern and Mid-Miocene molar samples with indication of elemental transects chosen for each section (red arrows). Note the inwards-outwards direction of all transects, along growth direction Specimen FCT-DCT 4945 in the top right, FCT-DCT 4933 bottom center and FCT-DCT 4943 on the far right. The modern elephant molar is depicted in the bottom left.



**Figure 5_Coimbra et al.**

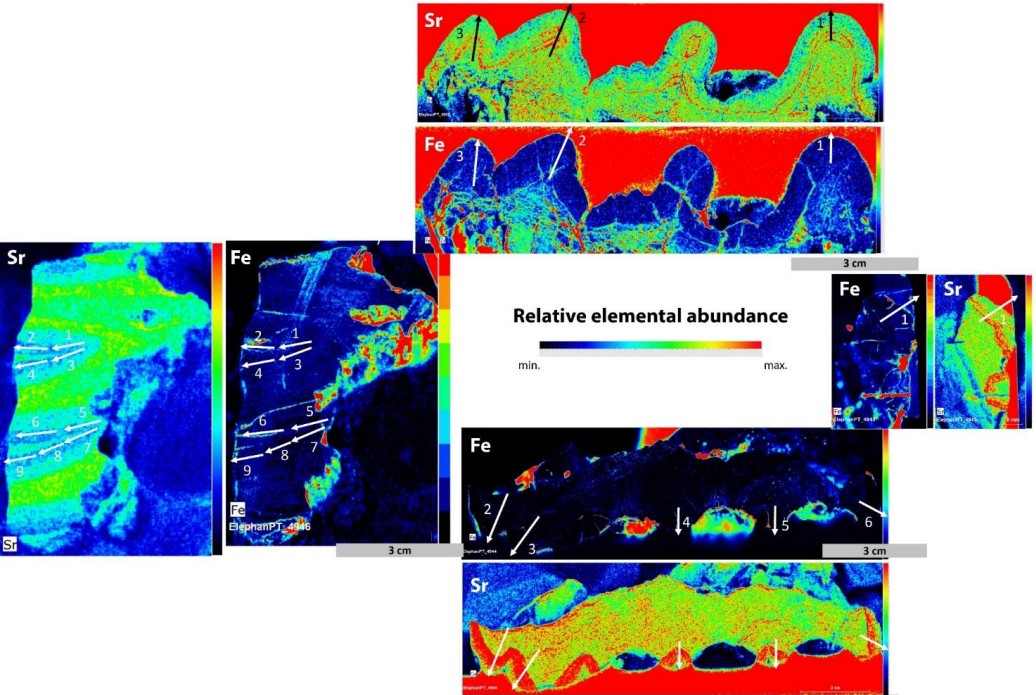

**Figure 5.** Relative elemental abundance for all specimen, here shown for iron and strontium (distribution of other elements can be found at https://doi.org/10.5281/zenodo.14882824). Note the lack of areas denoting higher iron abundance along selected transects and the clear Sr concentration pattern highlighting well-preserved tooth morphology.




**Figure 6_Coimbra et al.**

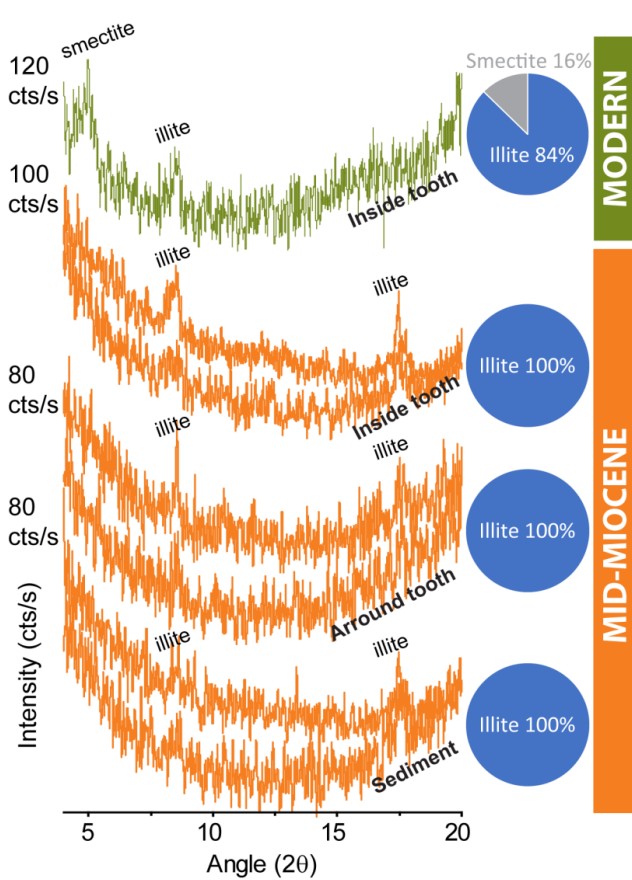

**Figure 6.** Clay mineral assemblage for Mid-Miocene sediment samples collected from areas adjacent to the molars, around and inside cavities. Modern sediment sample reflects soil substrate from the Lisbon Zoo (see text for details). Note the dominance of illite in all cases, along with minor contribution of smectite for the modern sample.





**Figure 7_Coimbra et al.**

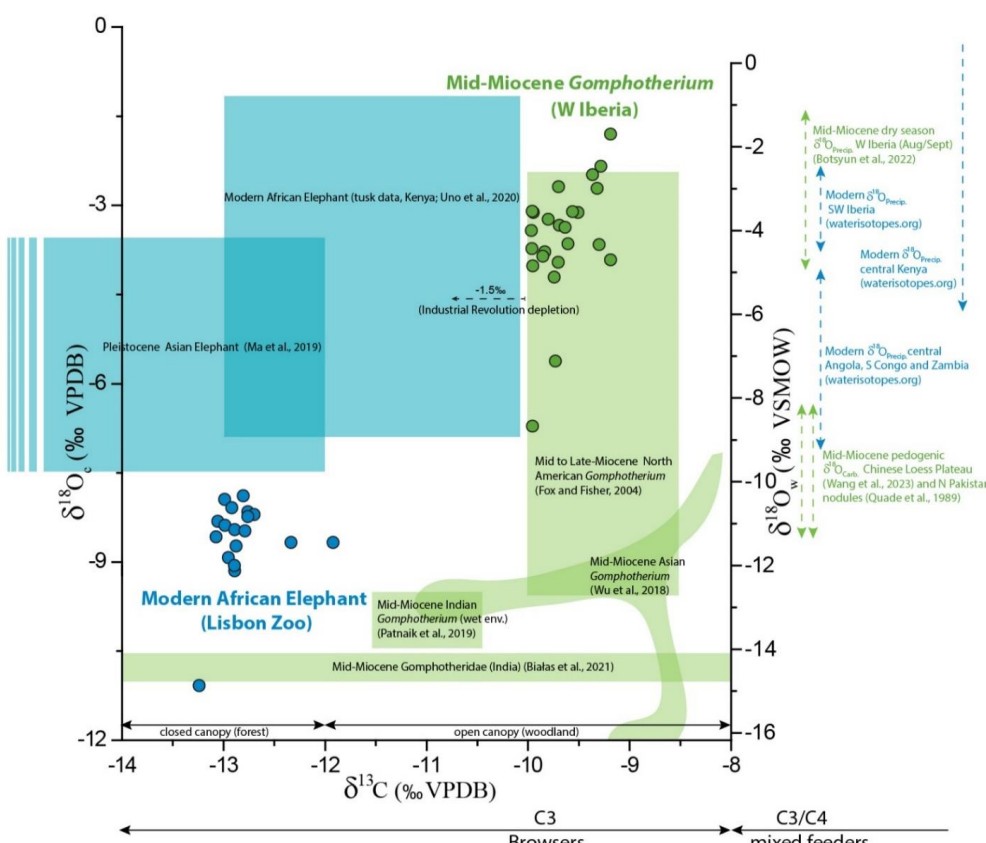

**Figure 7.** Biplot representation of stable carbon and oxygen isotope values, highlighting the differences between the two sets of samples tested (modern *vs.* Mid-Miocene). Shaded fields represent C and O isotope values in enamel-bound carbonate reported by other authors (see reference list) for Mid-Miocene and modern settings. Vertical arrows indicate range of variation of oxygen isotope values representative of rainfall composition at different localities (modern and ancient). Oxygen isotope values of water (right vertical scale) are compared to oxygen isotope values in bioapatite using the formula for large, water-dependent herbivores from (Hoppe, 2006) and we aligned the VSMOW to VPDB scale using the formula in (Coplen et al., 1983) following the methodology in (de Rooij et al., 2022) and (Wooller et al., 2021) . Forest/woodlands range of carbon-isotope variation after Patnaik et al. (2019) and references therein.



**Figure 8_Coimbra et al.**

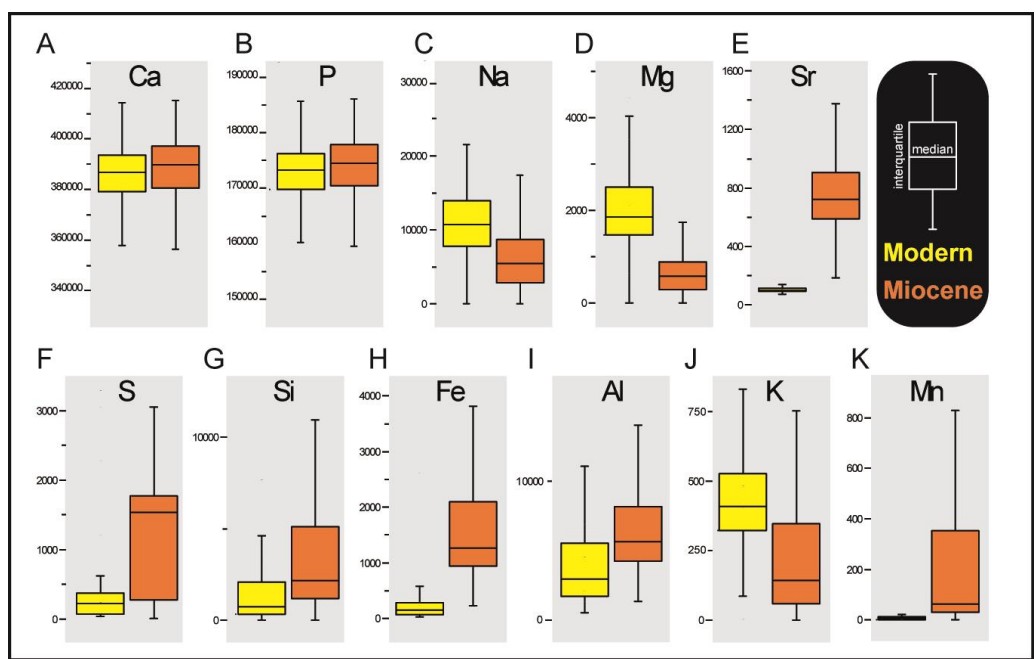

**Figure 8.** Boxplot representation (median and interquartile distance) of absolute elemental concentrations (in ppm). A and B) note similar distribution of Ca and P; C and D) lowered values for ancient samples regarding Na and Mg; E to K) overall higher median values for the remaining proxies (except K). See Table 2 for full elemental range, mean and standard deviation.





**Figure 9_Coimbra et al.**

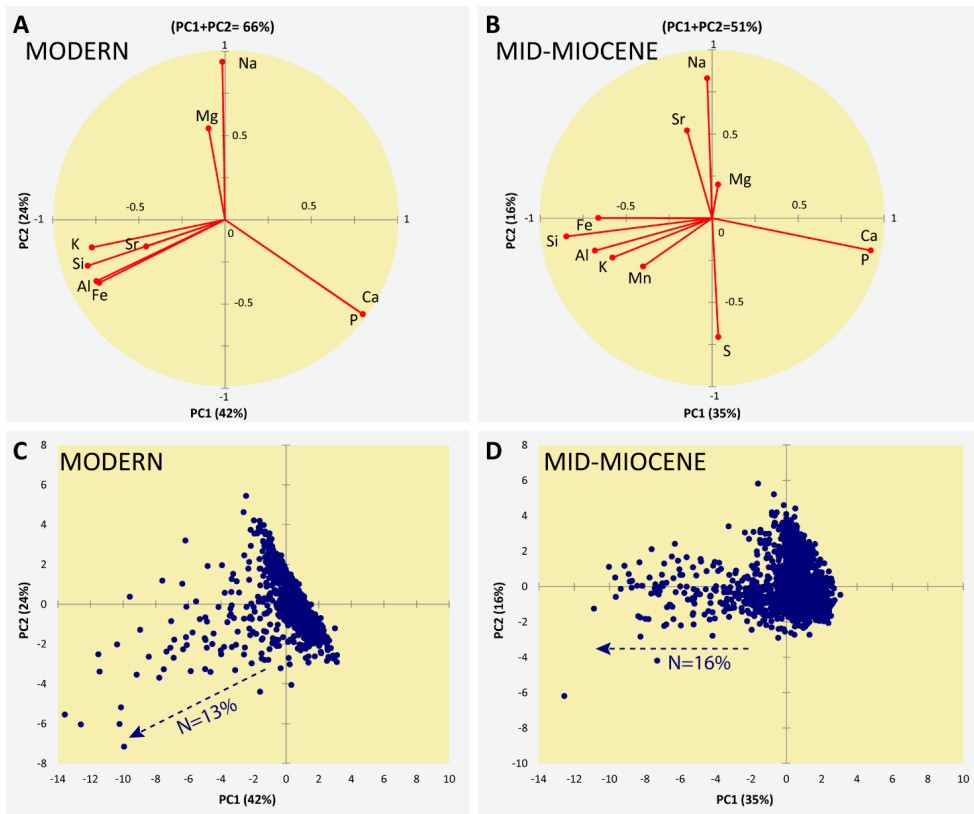

**Figure 9.** Principal component analysis computed for the modern and Mid-Miocene samples. A and B) Principal component loadings explaining 51 to 66% of the total variability of each set of samples. Note similar distribution of elements along the PC axis. Modern Mn and S were not used due to low number of measurements available (N<30%). C and D) Principal component scores indicating sample distribution along the PCA space. Note main clusters of samples in both cases, with only few samples (13% to 16% of the total number of samples) departing from the observed main trend.



**Figure 10_Coimbra et al.**

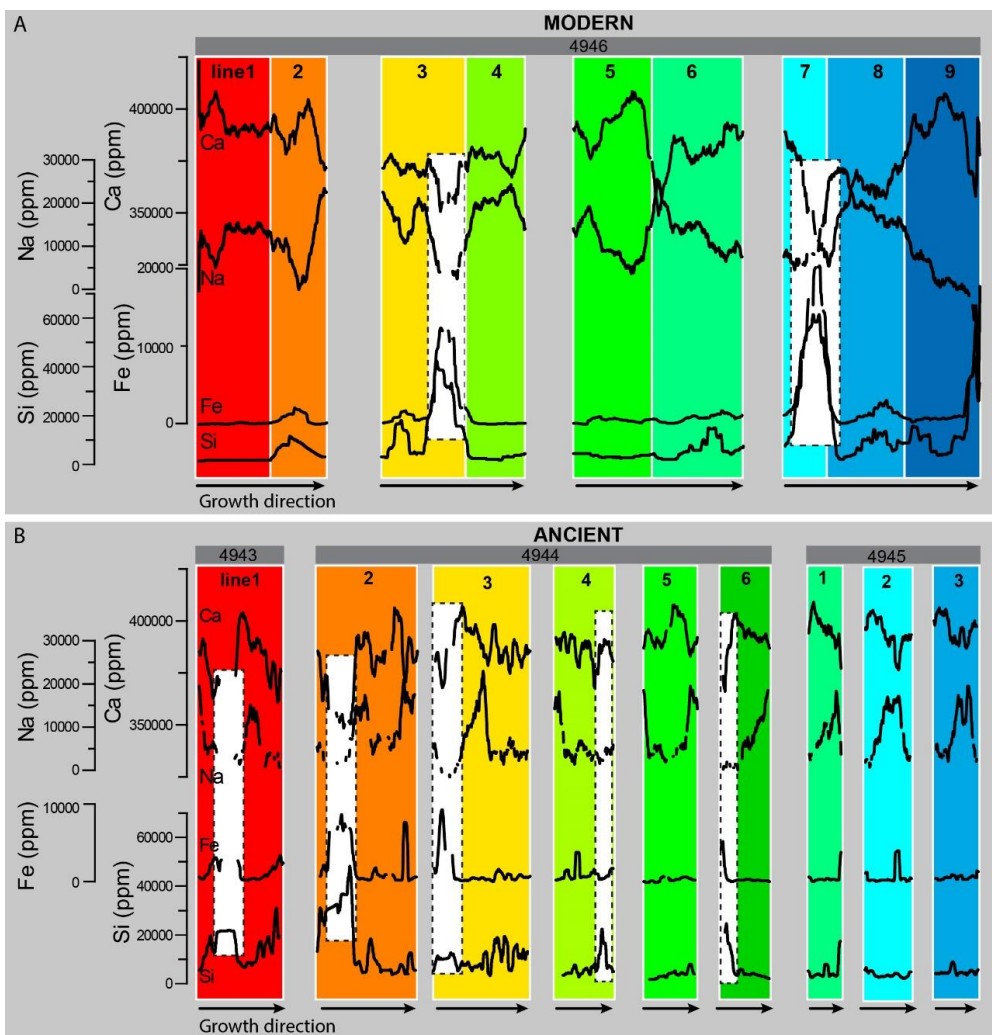

**Figure 10.** Elemental trend of selected proxies (Ca, Na, Fe and Si) for all measured transects (see Figs. 4 and 5 for exact measurement sites). A) Trend lines for modern transects. B) Trend lines for Mid-Miocene transects. Note overall similar records of Ca vs. Na and Fe vs. Si, as well as lack of clear enrichment or depletion trends along each transect. Note opposite variation pattern for Ca and Na, except when Fe and Si become significantly higher (white dashed boxes). Trend lines smoothed (10 points adjacent averaging); numerals according to Figure 4. Colors represent the different enamel transects analyzed. The same color scheme is used consistently across all figures to maintain clarity and allow for easy comparison of the different lines or transects.



**Figure 11_Coimbra et al.**

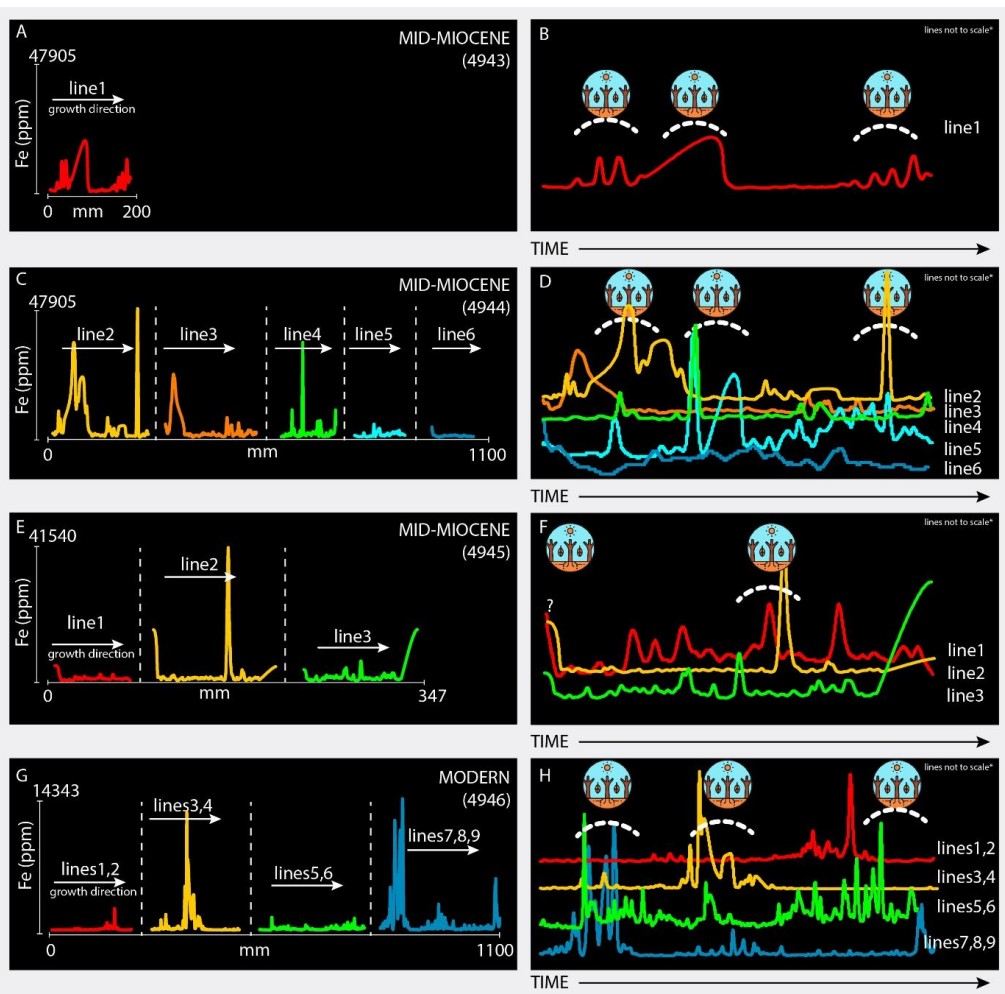

**Figure 11.** Elemental trend line obtained for Fe measurements (line numbers as in Figure 4). A, C E and G) Record of Fe abundance along growth direction (in mm). B, D F and H) Same datasets stacked and slightly adjusted (vertically and horizontally) to compare different transects that represent similar portions of the molar. (*) lines not to scale: mean horizontal scaling factor of 5.1; mean vertical scaling factor of 3.3 (see Table S1 in Appendix). Drawing represents dry season periods. Colors represent the different enamel transects analyzed. The same color scheme is used consistently across all figures to maintain clarity and allow for easy comparison of the different lines or transects.





**Figure 12_Coimbra et al.**

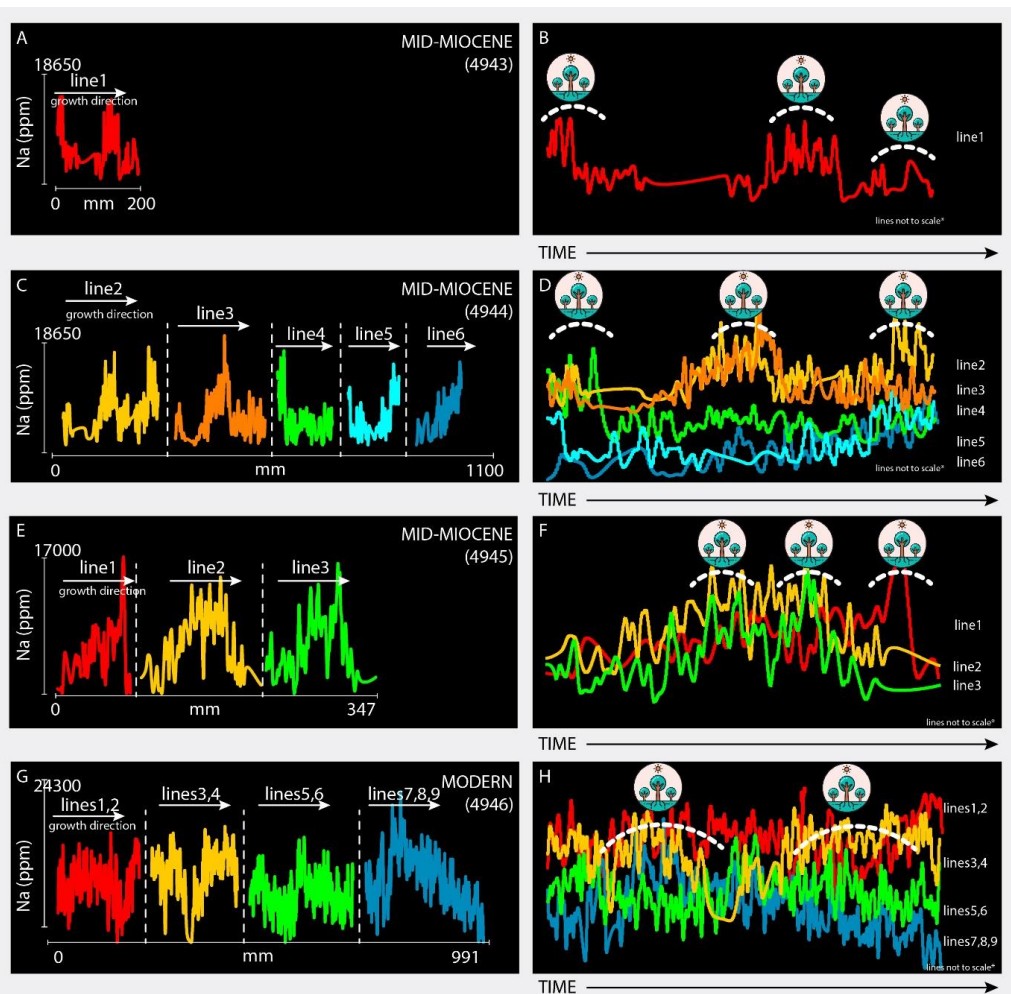

**Figure 12.** Elemental trend line obtained for Na measurements (line numbers as in Figure 3). A, C and E) Record of Na abundance along growth direction (in mm). B, D and F) Same datasets stacked and slightly adjusted (vertically and horizontally) to compare different transects that represent similar portions of the molar. (*) lines not to scale: mean horizontal scaling factor of 5.1; mean vertical scaling factor of 1 (see Table S1 in Appendix). Drawing symbol represents wet season periods. Colors represent the different enamel transects analyzed. The same color scheme is used consistently across all figures to maintain clarity and allow for easy comparison of the different lines or transects.




**Figure 13_Coimbra et al.**

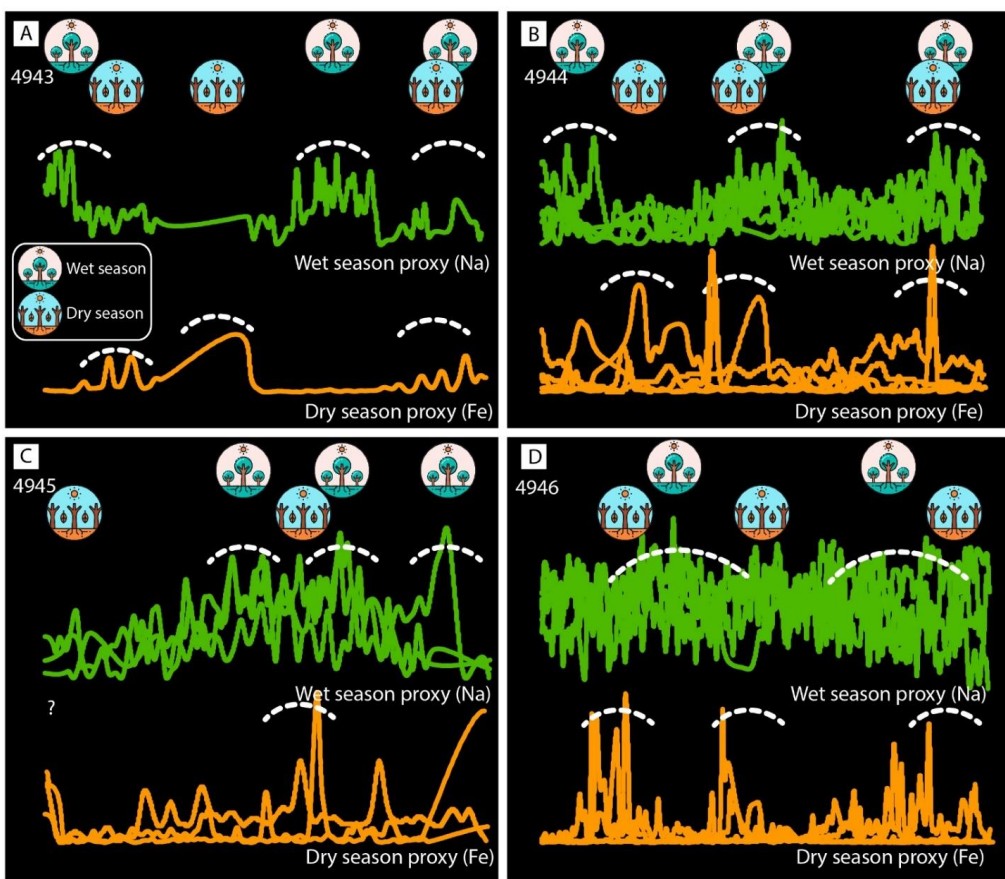

**Figure 13.** Compilation of elemental trend lines for Fe and Na as shown in Figures 11 and 12. A to C) Mid-Miocene transects in growth direction showing alternating peaks of higher abundance of the chosen elements. D) Modern transects in growth direction showing alternating peaks of higher abundance of the chosen elements.



Table 1. Semiquantitative abundance (%) of bulk mineralogical composition of sediment samples.

|  | Quartz | Calcite | Goethite | Ilmenite | Siderite | Dolomite | Observations |
|---|---|---|---|---|---|---|---|
| Sediment | 96 | 1 | 3 |  |  |  | yellow sediment |
|  | 94 | 2 | 4 |  |  |  | hard crust |
| Arround Tooth | 21 |  | 59 | 20 |  |  | yellow sediment |
|  | 32 |  | 41 |  |  | 27 | crust |
| Inside Tooth |  |  | 100 |  |  |  | red hard crust |
|  | 67 |  | 26 |  | 7 |  | yellow sediment |
| Zoo | 52 | 48 |  |  |  |  |  |

Table 2. Mean (±standard deviation; s.d.), minimum and maximum elemental values obtained for selected proxies obtained from Miocene and modern samples.

| Proxy | Mean | s.d. | min. | Max. | Proxy | Mean | s.d. | min. | Max. | Proxy | Mean | s.d. | min. | Max. |
|---|---|---|---|---|---|---|---|---|---|---|---|---|---|---|
| Ca | 387428 | 14136 | 331284 | 415258 | Na | 6042 | 3982 | 8 | 22222 | Si | 4450 | 5944 | 3 | 44398 |
|  | 385614 | 12089 | 331324 | 415589 |  | 10850 | 4459 | 2 | 26115 |  | 3278 | 7492 | 2 | 59156 |
| P | 173518 | 6331 | 148372 | 185982 | Mg | 645 | 465 | 1 | 4435 | Fe | 2621 | 4657 | 225 | 56135 |
|  | 172706 | 5414 | 148391 | 186130 |  | 2383 | 1831 | 2 | 15502 |  | 428 | 1357 | 26 | 18381 |
|  |  |  |  |  | Sr | 755 | 224 | 185 | 1510 | Al | 6891 | 4017 | 1332 | 25596 |
|  | MIOCENE |  |  |  |  | 107 | 34 | 63 | 722 |  | 4547 | 4825 | 512 | 45641 |
|  | MODERN |  |  |  | S | 1215 | 755 | 1 | 3046 | K | 403 | 701 | 0 | 5636 |
|  |  |  |  |  |  | 246 | 164 | 40 | 617 |  | 485 | 342 | 85 | 4276 |
|  |  |  |  |  |  |  |  |  |  | Mn | 506 | 1391 | 0 | 26942 |
|  |  |  |  |  |  |  |  |  |  |  | 18 | 48 | 0 | 365 |