# Peer review of "Detection of dietary stress and geophagic behaviour forced by dry seasons in Miocene 3 Gomphotherium"

_EGUsphere, 2025_

## Referee Comment (RC2)

[revised manuscript text omitted]

| Proxy | Mean              | s.d.  | min.   | Max.   | Proxy    | Mean  | s.d.     | min. | Max.  | Proxy | Mean | s.d. | min.  | Max.  |
|-------|-------------------|-------|--------|--------|----------|-------|----------|------|-------|-------|------|------|-------|-------|
| Ca    | 387428            | 14136 | 331284 | 415258 | Na       | 6042  | 3982     | 8    | 22222 | Si    | 4450 | 5944 | 3     | 44398 |
|       | 385614            | 12089 | 331324 | 415589 |          | 10850 | 4459     | 2    | 26115 |       | 3278 | 7492 | 2     | 59156 |
| P     | 173518            | 6331  | 148372 | 185982 | Μσ       | 645   | 465      | 1    | 4435  | Fe    | 2621 | 4657 | 225   | 56135 |
|       | 172706            | 5414  | 148391 | 186130 |          | 2383  | 1831     | 2    | 15502 |       | 428  | 1357 | 26    | 18381 |
|       |                   |       |        |        | 755      | 224   | 185      | 1510 | 4.7   | 6891  | 4017 | 1332 | 25596 |       |
|       | MIOCENE
MODERN |       |        | Sr     | 107      | 34    | 63       | 722  | Al    | 4547  | 4825 | 512  | 45641 |       |
|       |                   |       |        | S      | 1215     | 755   | 1        | 3046 | K     | 403   | 701  | 0    | 5636  |       |
|       |                   |       |        |        | 246      | 164   | 40       | 617  |       | 485   | 342  | 85   | 4276  |       |
|       |                   |       |        |        | - |       |   | •    | М     | 506   | 1391 | 0    | 26942 |       |
|       |                   |       |        |        |          |       |          |      |       | Mn    | 18   | 48   | 0     | 365   |

---

## Author Comment (AC1)

**Author's answer to comments**

"Detection of dietary stress and geophagic behaviour forced by dry seasons in Miocene *Gomphotherium*" egusphere-2025-1770

We very much appreciate the time and effort of both reviewers, the feedback provided is valuable to improve the manuscript. Please find below our point-by-point reply. Simple (linguistic) or very minor changes will be accommodated in the revised manuscript. Based on the comments made, we intend to clarify some portions of the text and introduce new supporting text.

**REVIEWER#1**

It would be beneficial to the reader access to more raw data in better-organized tables (or in the appendices depending on how much data is available).

It seems that this reviewer was unable to access the vast dataset available as online repository (<a href="https://doi.org/10.5281/zenodo.14882824">https://doi.org/10.5281/zenodo.14882824</a>), as well as the complementary details included in the Supplementary File. Reviewer #2 does not report such constraints ("Link works well. Good data backup). We believe most of the doubts laid by Reviewer#1 could have been clarified if this had not been a limitation. In any case, we provide clarifications to all comments below.

I did find myself asking many questions related to data I could not locate and I lay out these and other specific questions below:

**Section 1.1:** It is safe to presume *Gomphotherium* filled a similar niche as modern African elephants. What is that important in terms of climate change? *I.e., Why should we care how modern elephants might respond to climate change?* (I think there are a lot of cool ways to answer this!)

This is a very interesting suggestion. We will add a few lines to this topic, citing the relevant literature. We follow the reasoning that Gomphotheres likely shaped their environments much like modern elephants do, so their history gives us a window into what happens when a major ecosystem-engineer faces rapid change. If elephants today struggle with warming, drought, or shifting vegetation, their ecosystems will be impacted. They disperse seeds, open habitats, keep woody cover in check, and move nutrients, processes that strongly affect biodiversity and even carbon cycling. When past megaherbivores declined, those functions vanished, and landscapes became less resilient. So we care because elephants aren't just animals in the system, they are part of the system's stability. How they respond to climate change will influence the resilience of whole ecosystems, just as losing gomphotheres once did.

**Section 1.3, line 117:** Just looking for some clarification here: Can one modern proboscidean tooth capture 15 years?

Yes, we will clarify this by adding more pertinent references. According to research on enamel accretion rate (Dirks et al., 2012; Metcalfe and Longstaffe, 2012; Esker et al., 2019; Kowalik et al., 2023) reaching up to 13mm/yr, there is a possibility of capturing an environmental record spanning up to 15 years. Nevertheless, since molars get worn during

feeding, this span can significantly decrease (see examples illustrating such difference in Fig.3 A and C).

And would this imply that less time is captured in low-crowned gomphothere molars? This is an interesting point of view, we will add some text clarifying this aspect. Ultimately, the time captured in a single tooth is governed primarily by enamel extension rate and crown formation time. Crown height plays a secondary role, because wear can remove large portions of the developmental sequence regardless of morphology. Thus, low-crowned gomphothere molars may capture a shorter interval only if their enamel grew more slowly or if wear removed a larger proportion of the crown.

Based on the previous, in realistic terms for a low-crowned gomphothere molar (where preserved enamel lengths are often on the lower side, say  $\sim 30-80$  mm), one should expect on the order of  $\sim 2-8$  (up to  $\sim 15$ ) years of record in many cases, up to  $\sim 15$  years is plausible if the enamel record is long and accretion rate is high. But short (a few years) is equally plausible if enamel record is short or heavily worn.

**Section 1.3:** Is there a reason we don't look at the isotopic record in tusks? Does it have to do with how they grow?

Tusk enamel has been used to perform paleoenvironmental analysis (Fox and Fisher, 2001, Fox and Fisher, 2004). But legal constraints to acquire/transport tusks are a limitation due to strict anti-poaching laws, potentially limiting their use for research. Therefore, we relied on the available enamel material in this study.

**Section 1.5, lines 169-170:** C4 vegetation existed long before 8-5 Mya. Do you mean that it wasn't established the *study area* until 8-5 Mya?

We will clarify in the revision that the wide spread of C4 vegetation in our area dates to this time. C4 grasses originated well before the late Miocene, but what Cerling et al. (1997) describes is not their first appearance, it is the major ecological expansion of C4 biomass between ~8 and 6 Ma. This expansion was highly heterogeneous across latitudes: lower-latitude regions reached C4 dominance earlier because warmer temperatures shift the threshold at which C4 plants outcompete C3 taxa. Thus, while C4 vegetation certainly existed earlier at the global scale, its substantial increase in relative abundance likely occurred later in our study area. We will clarify this distinction between origin, ecological dominance, and regional timing in the revised manuscript.

**Section 1.5, lines 179-181:** Are you implying that the present-day climate of the Guinea Gulf is tropical?**

We will support this information with references and mention Köppen classification to the revised version. The Gulf of Guinea has a tropical monsoon climate (Köppen classification "Am") characterized by high temperatures, high humidity, and heavy rainfall year-round, driven by the Intertropical Convergence Zone (ITCZ) and the West African monsoon. There is a main rainy season from April to October and a drier season from November to March, though squalls with rain can occur at any time. Average temperatures are generally high and vary little seasonally, though they decrease with altitude in inland regions.

And were the sample sites still so close to the water during the Miocene (was the paleogeography similar)?

Miocene paleogeography for the Iberian Peninsula was very similar to modern configuration, as recently documented by He et al. (2023), their Figures 2 and 3. We will add this information to the revised manuscript, highlighting this point.

He, Z., Zhang, Z., Guo, Z., Scotese, C. R. & Deng, C., 2023. An early miocene (~20 ma) paleogeographic reconstruction for paleoclimate modelling. Palaeogeography, Palaeoclimatology, Palaeoecology, 612(111382).

Section 1.5, lines 183: "Subsequently?" This seems right in the middle of the Burdigalian and Langhian stages. I think this entire paragraph is a bit confusing as written. Consider revising.

We thank the reviewer for pointing out this confusion. On re-reading the paragraph, we agree with their point. We will revise this paragraph and pay special attention to the chronological and dating statements.

**Section 2.1:** Is it possible that the gomphothere molars belong to juveniles (how can you rule this out?). The inclusion of juvenile teeth could have implications for stable isotope interpretations.

We agree this is a relevant point to be clarified in the revised version, so new text will be added on this topic. We can rule out a juvenile origin for these molars based on several independent indicators. First, the teeth show fully developed crown morphology and plate count appropriate for adult stages; juvenile gomphotheres have fewer lamellae, reduced crown height, and incomplete enamel formation. Second, the wear facets indicate sustained adult mastication rather than the light or uneven wear typical of juveniles. Third, the dimensions of the preserved molars fall within adult size ranges for the taxon. Taken together, these features are incompatible with a juvenile ontogenetic stage, so juvenile bias does not affect our isotopic interpretations.

Section 3.1: Playing devil's advocate here, but are three gomphothere molars and one modern elephant molar enough to base your conclusions upon? Might be worth a sentence explaining the significance/rarity of these specimens for any non-paleontologist readers. The reviewer is absolutely right. Reviewer#2 also mentions this aspect. We will clarify in our revised discussion that our conclusions have limitations due to the limited numbers of individuals they are based on, and highlight the difficulty in obtaining material like this to motivate our study design earlier in the manuscript. Although our sample comprises only three gomphothere molars and one modern elephant molar, these specimens are exceptionally rare and well-preserved, providing high-resolution geochemical records that are otherwise unavailable for these taxa. Even a small number of teeth can yield meaningful insights into long-term dietary patterns and environmental conditions, particularly when each tooth captures several years of enamel formation. The scarcity of such material underscores the significance of these analyses for understanding proboscidean paleoecology.

**Section 4.1, line 328:** Which teeth are you referring to here, all three? Or are these abundances from various sample locations along a single tooth? Please clarify.

This information will be added to the Methods section, clarifying the materials used for XRD analysis. One specimen was chosen for XRD analysis: sample 4943 (Figs. 3 and 4).

**Section 4.1, TABLE 1:** Since this is only three gomphothere molars, it is extremely important to see a more detailed and better organized table that actually lays out the data values for each specimen. If it is a lot of data, this could go into an appendix.

Unfortunately, it appears the reviewer could not access our digital data repository. All data used in this contribution are detailed there. We invite the reviewer to access it, if they are invited for a second review round, and welcome any feedback on the organization of our data in that online supplement.

Section 4.3, lines 396-397: Sentence here is confusing as written. Please reword.

Yes, we agree this sentence needs rephrasing, as follows:

"Since diagenetic alteration can lead to depletion or enrichment of some elements (Fig. 8 and Table 2), we focus on discussing elemental trends and associations (rather than absolute abundance), as these features may enclose (paleo) environmental fluctuations during the growing period (typically <4/5years (Uno et al., 2020)."

**Section 5.2.1:** This entire section seems a bit out of place here. This information seems like it should be included in section 1.2.

Placing this information here served as a small introduction to the following text. But it can be moved up to section 1.2 to fit the structure of the article. We are happy to follow the reviewer's suggestion and move this to section 1.2 in our revised manuscript.

**Section 5.2.1, lines 525-527:** Enamel biomineral composition responds to the physiological and taxonomical characteristics of the environment? Please clarify.

Our original sentence does not match the idea stated by the reviewer. To avoid further confusion, we will add some clarification (underlined text): "During precipitation, enamel biomineral composition not only respond to the local chemical environment, i.e. the bioavailability of chemical elements, but also to physiological and taxonomical characteristics. The latter is clear from species-specific variations in biomineral composition in taxa that occur in the same environment."

**OTHER IMPORTANT COMMENTS:**

The following comments are clarified by consulting the digital repository:

- An appendix is referred to in some of the figure captions, but I cannot locate it (no appendix in supplementary data file).
- The metadata xml file is inaccessible (some sort of error).
- Please provide a data table of stable isotopes (C and O)!

• Did authors perform a seasonal analysis of d13C and d18O within the teeth?

No, seasonal trends were not tested for C and O stable isotopes.

Again, a data table that clarifies this would be helpful, but it would also be interesting to see this isotope data laid out similar to figures 11-13.

Our full isotope dataset is available online. Since our goal was not to provide seasonal variations for C and O stable isotopes, mimicking data presentation of figures 11 to 13 is not necessary in our opinion.

**REVIEWER#2 (and PDF)**

More-than-minor comments collected from the PDF file:

- the term "geophagic" may not be widely understood by the readers. I also realize that you define it below;

We believe that this term best captures what we mean in terms of behavior in gomphotheres so we prefer to keep using it. Additionally, we believe our definition is clear enough for the reader to follow our discussion.

- The interpretation of terrestrial d18O signals preserved in bioapatite is complex. N. Levin et al. (in refs.) also argue that this may represent a proxy for aridity in terrestrial ecosystems. Might want to also quote her work here.

Yes, this is certainly a good recommendation. We will add reference to this useful work in our revised manuscript where we explain the significance of the  $\delta^{18}O$  as a proxy for paleoenvironmental change.

- "No pre-treatment is carried out since the samples were obtained from clean cross sections through the molars and were considered unlikely to have been in contact with diagenetic fluids considering the taphonomic history of the sample site (see section 2)." This statement needs further support to be considered a valid argument.

Yes, we agree that more information can be added to support this procedure. We therefore plan to repeat some of the evidence for the taphonomic history of the specimens from section 2 here, so the reader can more easily follow our reasoning here.

- "possible origins of the modern specimen include central Angola, southern Democratic Republic of the Congo or Zambia, where the more depleted range of surface water  $\delta 180$  values overlaps with  $\delta 180$  values of drinking water expected based on the enamel- $\delta 180$  values in our modern elephant specimen ("Water Isotopes Database," 2023)". Interesting interpretation, but perhaps too speculative.

Due to limited documentation, we cannot know with certainty which elephant the molar came from. A best guess can be based on geochemical data and by comparing this data to literature. Certainly, Lisbon water sources can be ruled out, so the search for potential origin may be speculative, but still exciting to learn so much from the available dataset. We will add a comment on this in the revised version of the manuscript.

My primary concerns about this paper include the following:

- The small sample size (3 fossil teeth, 1 modern tooth) upon which this research study is based, and the interpretations that follow from them.

Reviewer #1 also mentioned this aspect. Therefore, we will add some discussion in our revised version clarifying the rarity and value of the presented record and exploring potential limitations (see answer to questions of Review # 1).

- The interpretation of the d18O signatures, which are potentially more complex than the authors indicate.

Yes, we agree aridity in terrestrial ecosystems is also a factor to take into account. We will include the helpful reference suggested above to add a bit more discussion on the impact of aridity on our  $\delta^{18}$ O results.

(3) The conclusions and comparisons of the fossil versus modern data are not as clearly integrated and argued as they should be.

We will do our best to integrate data from both sources (ancient and modern) in a more evident way, adding specific comments during data presentation and elaborating on similarities/differences along the corresponding discussion.